# Global analysis of cell behavior and protein dynamics reveals region-specific roles for Shroom3 and N-cadherin during neural tube closure

**Austin T Baldwin, Juliana H Kim, Hyemin Seo, John B Wallingford***

Department of Molecular Biosciences, University of Texas at Austin, Austin, United States

**Abstract** Failures of neural tube closure are common and serious birth defects, yet we have a poor understanding of the interaction of genetics and cell biology during neural tube closure. Additionally, mutations that cause neural tube defects (NTDs) tend to affect anterior or posterior regions of the neural tube but rarely both, indicating a regional specificity to NTD genetics. To better understand the regional specificity of cell behaviors during neural tube closure, we analyzed the dynamic localization of actin and N-cadherin via high-resolution tissue-level time-lapse microscopy during *Xenopus* neural tube closure. To investigate the regionality of gene function, we generated mosaic mutations in *shroom3*, a key regulator or neural tube closure. This new analytical approach elucidates several differences between cell behaviors during cranial/anterior and spinal/posterior neural tube closure, provides mechanistic insight into the function of *shroom3*, and demonstrates the ability of tissue-level imaging and analysis to generate cell biological mechanistic insights into neural tube closure.

**\*For correspondence:** wallingford@austin.utexas.edu

**Competing interest:** The authors declare that no competing interests exist.

## Editor's evaluation

This manuscript by Baldwin and colleagues on vertebrate neural tube closure will be of interest to developmental and cell biologists studying tissue morphogenesis as well as human geneticists focusing on neural tube defects. It is timely, as it introduces a new technology for large-scale imaging of cell behaviours in large embryos. Specifically, it uses advanced image analysis to quantitatively describe and correlate active cell behaviours and localization dynamics of key cytoskeletal and adhesion proteins driving a central step of neural tube closure. Data analysis is detailed and followed by careful conclusions.

## Introduction

Congenital birth defects are the number one biological cause of death for children in the US, and neural tube defects (NTDs) represent the second most common class of human birth defect (*Murphy et al., 2018*; *Wallingford et al., 2013*). NTDs represent a highly heterogenous group of congenital defects in which failure of the neural folds to elevate or fuse results in a failure of the skull or spine to enclose the central nervous system (*Wallingford et al., 2013*). While genetic analyses in both humans and animal models have revealed dozens of genes necessary for normal neural tube closure, several key questions remain.

One central unanswered question relates to the regional heterogeneity of both normal neural tube closure and pathological NTDs. For example, the collective cell movements of convergent

extension dramatically elongate the hindbrain and spinal cord of vertebrates, but not the midbrain and forebrain (*Nikolopoulou et al., 2017*; *Wallingford et al., 2013*). Accordingly, disruption of genetic regulators of convergent extension such as the planar cell polarity (PCP) genes results in failure of neural tube closure in posterior regions of the neural ectoderm, but not anterior (*Kibar et al., 2001*; *Wang et al., 2006*). Conversely, the *shroom3* gene is implicated in apical constriction, a distinct cell behavior that drives epithelial sheet bending, and disruption of *shroom3* elicits highly penetrant defects in anterior neural tube closure, but only weakly penetrant defects in the posterior (*Haigo et al., 2003*; *Hildebrand and Soriano, 1999*). This regional deployment of apical constriction in the anterior and convergent extension in the posterior during neural tube closure is poorly understood.

In addition, the underlying mechanisms of individual cell behaviors necessary for neural tube closure remain incompletely defined. While apical constriction is driven by actomyosin contraction, the precise site of actomyosin action during this process is unclear and constitutes a long-term problem in the field (*Martin and Goldstein, 2014*). For example, analysis of apical constriction during gastrulation in both *Drosophila* and *Caenorhabditis elegans* has shown integration of discrete junctional and medio-apical ('medial') populations of actomyosin (*Coravos and Martin, 2016*; *Martin et al., 2009*; *Roh-Johnson et al., 2012*). Recent studies in frog and chick embryos have also described similar pulsed medial actomyosin-based contractions occurring during neural tube closure (*Brown and García-García, 2018*; *Christodoulou and Skourides, 2015*; *Suzuki et al., 2017*), but how those contractions are controlled and how they contribute to tissue-wide cell shape changes during neural tube closure are not known.

For example, Shroom3 is among the more well-defined regulators of apical constriction, being both necessary and sufficient to drive this cell shape change in a variety of cell types, including the closing neural tube (*Haigo et al., 2003*; *Hildebrand, 2005*; *Plageman et al., 2010*; *Plageman et al., 2011b*). Shroom3 is known to act via Rho kinase to drive apical actin assembly and myosin contraction (*Das et al., 2014*; *Hildebrand, 2005*; *Nishimura and Takeichi, 2008*; *Plageman et al., 2011a*). However, the relationships between Shroom3 and the medial and junctional populations of actin have not been explored.

An additional outstanding question relates to the interplay of actomyosin contractility and cell adhesion during apical constriction. The classical cadherin Cdh2 (N-cadherin) is essential for apical constriction during neural tube closure in *Xenopus* (*Morita et al., 2010*; *Nandadasa et al., 2009*), and *shroom3* displays robust genetic interactions with *n-cadherin* in multiple developmental processes, including neural tube closure (*Plageman et al., 2011b*). Moreover, a dominant-negative N-cadherin can disrupt the ability of ectopically expressed Shroom3 to induce apical constriction in MDCK cells (*Lang et al., 2014*). Nonetheless, it is unclear if or how Shroom3 controls the interplay of N-cadherin and actomyosin during apical constriction. This is an important gap in our knowledge, because despite the tacit assumption that cadherins interact with each other and control actomyosin at cell-cell junctions, N-cadherin displays multiple cell-autonomous activities (*Rebman et al., 2016*; *Sabatini et al., 2011*). Intriguingly, several papers now demonstrate that extra-junctional cadherins at free cell membranes can engage and regulate the actomyosin cortex (*Ichikawa et al., 2020*; *Padmanabhan et al., 2017*; *Sako et al., 1998*; *Wu et al., 2015*).

Finally, though Shroom3 has been extensively studied in the context of cranial apical constriction, the gene is expressed throughout the neural plate (*Haigo et al., 2003*; *Hildebrand and Soriano, 1999*) and recent studies have also implicated Shroom family proteins in the control of convergent extension (*McGreevy et al., 2015*; *Nishimura and Takeichi, 2008*; *Simões et al., 2014*). Several studies indicate a genetic and cell biological interplay of Shroom3 and the PCP proteins (*Durbin et al., 2020*; *McGreevy et al., 2015*), and one study directly links PCP, apical constriction, and convergent extension (*Nishimura et al., 2012*). Conversely, some studies also suggest a role for PCP proteins in apical constriction (*Ossipova et al., 2015*).

Together, these studies highlight the complexity of neural tube closure, which is compounded by the sheer scale of the tissue involved. The neural ectoderm is comprised of hundreds to thousands of cells (depending on organism) and stretches from the anterior to posterior poles of the developing embryo. However, the vast majority of dynamic studies of cell behavior in the neural tube closure, including our own, have focused on small numbers of cells due to constraints of both imaging and image analysis.

Here, we used image-tiling time-lapse confocal microscopy to obtain over 750,000 individual measurements of cell behaviors associated with neural tube closure in *Xenopus tropicalis*. Using these data, we demonstrate that the cell biological basis of apical constriction differs substantially between the anterior and the posterior neural plate. The data further suggest that the crux of Shroom3 function lies not in actin assembly per se, but rather in the coupling of actin contraction to effective cell surface area reduction. Third, we demonstrate that the control of N-cadherin localization is a key feature of Shroom3 function during neural tube closure. Finally, we demonstrate that the incompletely penetrant posterior phenotypes related to *shroom3* loss stem from dysregulation of both actin and N-cadherin localization. Overall, these findings (a) elucidate differences between cell behaviors during cranial/anterior and spinal/posterior neural tube closure, (b) provide new insights into the function of *shroom3*, an essential neural tube closure gene, and (c) demonstrate the power of large-scale imaging and analysis to generate both cell-level mechanistic insight and new hypotheses for exploring neural tube closure.

## Results and discussion

### High-content imaging of cell behavior and protein localization during vertebrate neural tube closure

*X. tropicalis* affords several advantages for imaging neural tube closure, as its cells are large and easily accessible; its culture conditions for imaging are no more complex than synthetic pond water held at room temperature; and its broad molecular manipulability allows examination of diverse fluorescent markers. We developed methods for confocal microscopy and image tiling to collect high-magnification datasets spanning broad regions of the folding neural ectoderm from embryos injected at blastula stages with mRNAs encoding fluorescent reporters (*Figure 1A*). At the onset of neurulation (approximately *Nieuwkoop and Faber, 1994*; *Nieuwkoop and Faber, 1994*, stages 12.5–13), embryos were positioned to image either the anterior or the posterior regions of the neural ectoderm. We then established a pipeline by which cells captured in our movies were segmented using Tissue Analyzer, CSML, and EPySeg (*Aigouy et al., 2020*; *Aigouy et al., 2016*; *Ota et al., 2018*), yielding a map of both the apical cell surfaces and all individual junctions (*Figure 1B*). Finally, we built pipelines to process these data with Tissue Analyzer (*Aigouy et al., 2010*; *Aigouy et al., 2016*) and Fiji (*Schindelin et al., 2012*) to quantify both cell behaviors and the localization of fluorescent protein reporters across the neural plate and across neurulation.

With these methods in place, we considered three interrelated problems in neural tube closure biology: First, the incidence and form of NTDs differ widely between the brain and spinal cord (*Nikolopoulou et al., 2017*; *Wallingford et al., 2013*), yet our understanding of the dynamic cell behaviors in the two regions remains limited. Second, a unified mechanism for apical constriction has emerged in recent years involving the coordinated action of two discrete populations of actomyosin positioned either at apical cell-cell junctions or the medial apical cell surface (*Coravos and Martin, 2016*; *Martin and Goldstein, 2014*; *Martin et al., 2009*; *Roh-Johnson et al., 2012*), but the extent to which this model, developed in *Drosophila* and *C. elegans*, applies to vertebrates is unknown. Third, N-cadherin is essential for apical constriction in *Xenopus* (*Nandadasa et al., 2009*), but its functional interplay with junctional and/or medial actin is unknown. Accordingly, we made movies focused on either the anterior or posterior neural plate during neurulation, imaging the fluorescent actin biosensor LifeAct-RFP (*Riedl et al., 2008*; *Figure 1A*, magenta) and N-cadherin-GFP (*Figure 1A*), and independently quantified the mean fluorescent intensity of junctional and medial populations for both reporters (*Figure 1C, D*). To account for noise in these measurements, we have smoothed the data within individual cell tracks by averaging the data over a 7-frame window (*Figure 1—figure supplement 1A, B*).

In total, our dataset is comprised of ~250,000 observations of apical cell surfaces from over 3700 cells and ~580,000 observations from over 13,000 individual cell-cell junctions across nine embryos (*Figure 1D, E*). Images were collected at a rate of 1 frame/observation per minute over 1–2 hr, spanning roughly stages 13–18. Initial cell sizes and fluorescent varied among cells and embryos due to both natural variation and staging as well as variation introduced via mRNA microinjection. Because our primary interest is in the dynamics of apical constriction, we standardized many of the parameters in our analyses to account for variation in cell size and fluorescent intensity. This standardization

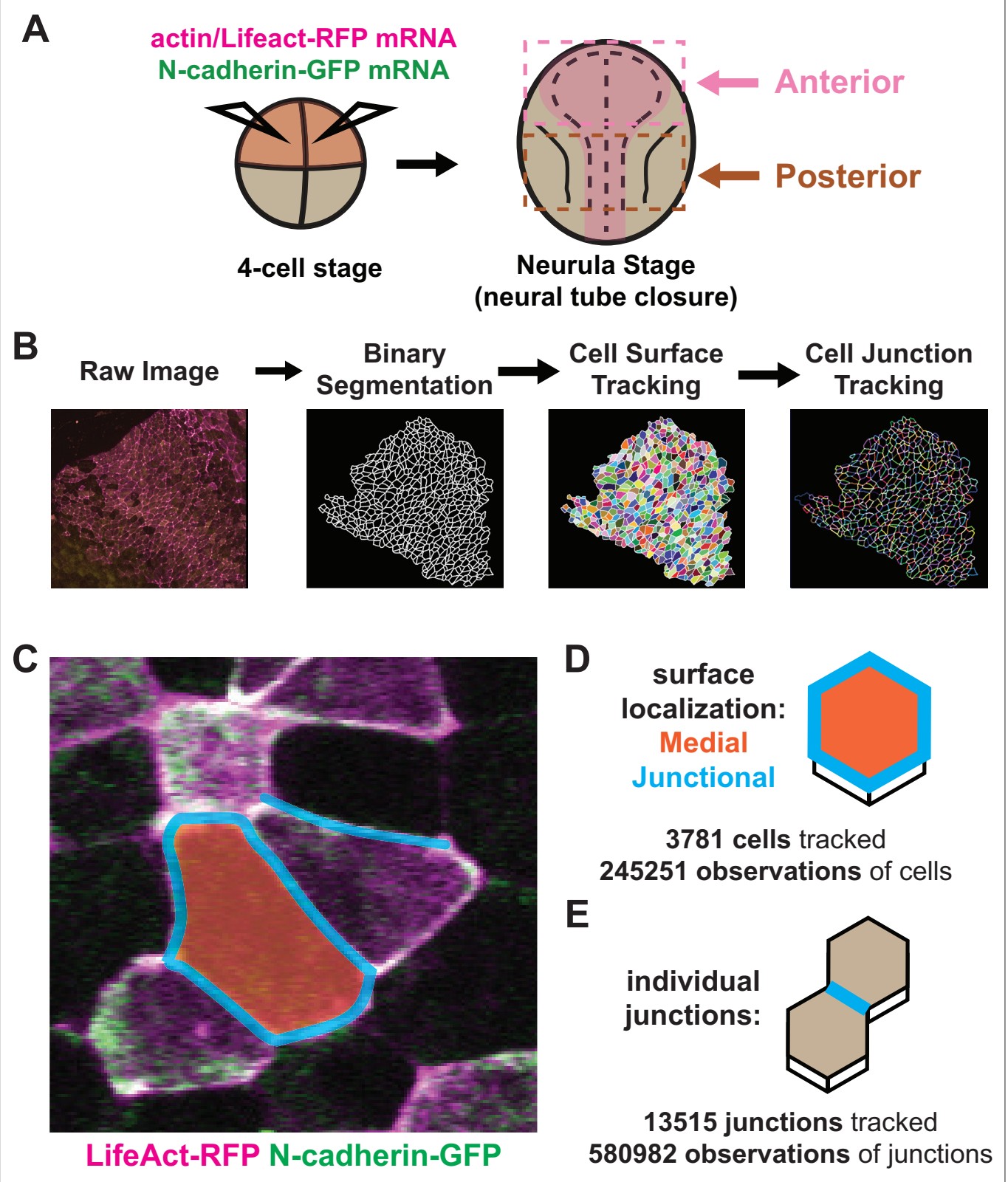

**Figure 1.** Tissue-level imaging and analysis of contractile protein dynamics during neural tube closure in *Xenopus*. (**A**) Schematic of mRNA injections and subsequent imaged regions of the *Xenopus tropicalis* embryo. (**B**) Cell segmentation and tracking workflow. Binary segmentation, cell surface tracking, and cell junction tracking were all generated using Tissue Analyzer. (**C**) Example *Xenopus* cells with analyzed subcellular domains labeled.

*Figure 1 continued on next page*

Figure 1 continued

Orange label = medial, cyan labels = junctional/junctions. (**D**) Schematic and N values of whole cell measurements. (**E**) Schematic and N values of individual cell junction measurements.

The online version of this article includes the following figure supplement(s) for figure 1:

**Figure supplement 1.** Per cell data processing and analysis.

involved mean-centering the data for each individual cell track to zero and then dividing the resulting mean-centered values by the standard deviation of each track, such that the standardized parameters are now measured in standard deviations rather than square microns or arbitrary units, allowing for simpler comparisons of overall changes in parameters between cells over time (*Figure 1—figure supplement 1A, D*).

We first performed an initial test of the validity of our approach, examining our dataset for well-known trends expected for neural epithelial cells during neural tube closure, namely an overall decrease in apical area and an overall increase in apical actin intensity. The heat maps in *Figure 2A, B* reveal that cells in both anterior and posterior regions generally reduce their apical surface area and increase medial actin intensity, as expected. These overall trends are backed by examination of individual cells, as shown for specific representative cells in *Figure 2C*. Overall, this analysis suggests that our pipeline is generally effective for quantifying cell shape and actin intensity over time during neural tube closure.

## Distinct patterns of apical constriction behavior, actin assembly, and N-cadherin localization in anterior and posterior regions of the closing neural tube

Our dataset revealed several interesting trends. First, while bulk measurements showed a decrease in apical area in both anterior and posterior regions over time (*Figure 3A*), we observed distinct region-specific distributions for these changes. For example, in the anterior, the vast majority of cells displayed significant apical constriction, and this constriction proceeded gradually across neurulation (*Figure 3A and A'*, left). In the posterior, however, a much smaller proportion of posterior cells constricted and a substantial number actually dilated (*Figure 3A*, right). Moreover, constriction of cells in the posterior was initiated very late in neurulation and proceeded very rapidly (*Figure 3A'*, right).

We further observed that both medial and junctional actin intensity generally increased over time in both regions, with their temporal progressions being reciprocal to the changes in apical area described above (*Figure 3B, B', C, and C'*, right). Again, these distributions were significantly different between the anterior and posterior regions, with cells in the spine having a significantly more heterogeneous distribution of actin accumulation outcomes (*Figure 3B, C*, left).

By far the most intriguing results related to the dynamics of N-cadherin localization, for which we observed two surprising patterns. First, in the anterior neural plate N-cadherin accumulated dramatically not only in the junctional region *but also in the medial* region (i.e. the free apical surface) (*Figure 3D, D', E and E'*). Thus, N-cadherin localization closely parallels actin dynamics in the normal anterior neural plate. This result was surprising because classical cadherins such as N-cadherin are typically known for their action at cell-cell junctions. Nonetheless, immunostaining for endogenous N-cadherin in fixed embryos confirmed this medial accumulation in the apical surfaces of anterior neural ectoderm cells (*Figure 3—figure supplement 1*). Our dataset lacked the time resolution to determine precise patterns of N-cadherin movement during apical constriction, but in Z-projections of highly constricted cells, we observed N-cadherin signal not just coincident with, but also basal to, to the apical actin signal (*Figure 4*). This result is consistent with the emerging understanding of the cell-autonomous roles for cadherins in both actin organization and endocytosis (*Ichikawa et al., 2020*; *Padmanabhan et al., 2017*; *Rebman et al., 2016*; *Sabatini et al., 2011*; *Sako et al., 1998*; *Wu et al., 2015*).

The medial accumulation of both N-cadherin and actin led us to explore the localization of Shroom3 itself. No antibodies are available for *Xenopus* Shroom3, and gain-of-function effects during early development preclude analysis of tagged wild-type Shroom3 during neural tube closure. That said, ectopic Shroom3 clearly decorates both junctional and medial regions in diverse epithelial cells (*Haigo et al., 2003*; *Kowalczyk et al., 2021*; *Lee et al., 2009*). To gain insight into Shroom3

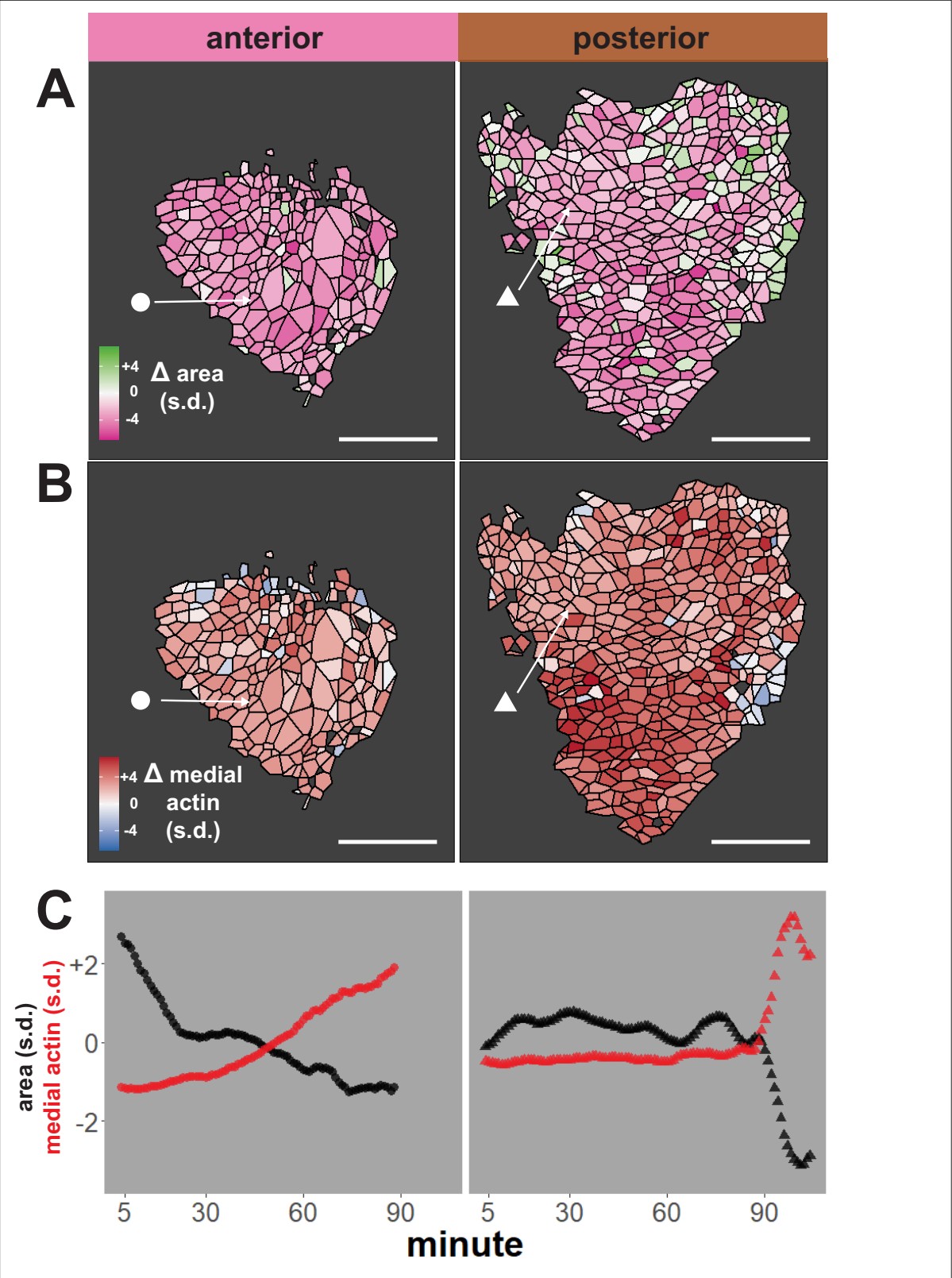

**Figure 2.** Tissue-level analysis of individual cell behaviors reveals dynamic heterogeneity. (**A**) Overall change (Δ) in apical surface area (standardized) across anterior (left) and posterior (right) control embryos. (**B**) Overall change in medial LifeAct/actin localization (standardized) across anterior (left) and posterior (right) control embryos. Circle and triangle in A and B denote a representative cell for each embryo. Scale bars = 100 μm. (**C**) Standardized apical surface area (black) and medial actin (red) over time in representative cells from anterior (left/circle) and posterior (right/triangle). s.d. = standard deviation.

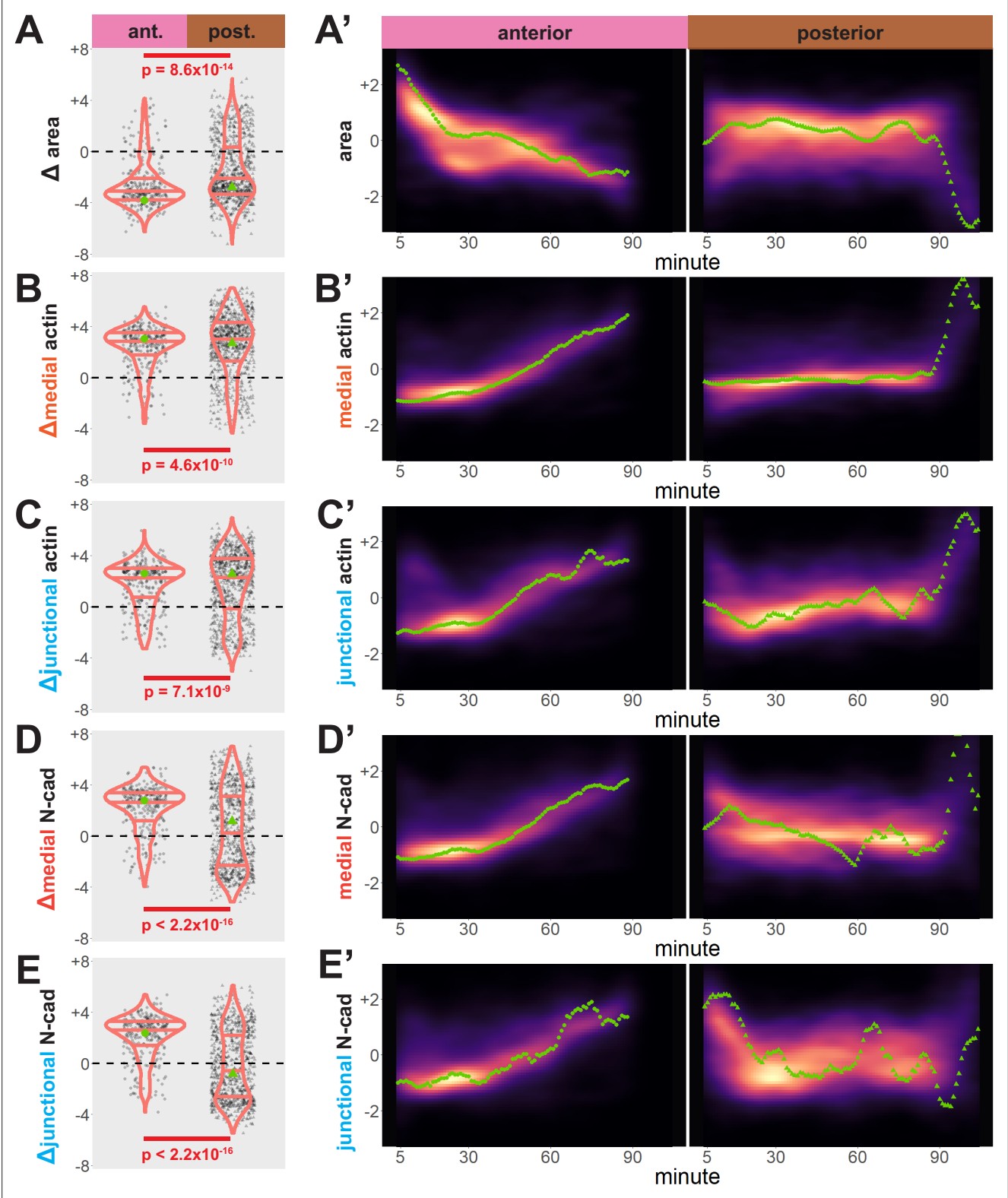

**Figure 3.** Cells in the anterior and posterior neural ectoderm both apically constrict but differ in their contractile protein dynamics. Tissue-level cell size and protein localization dynamics from control embryos in *Figure 2*. (X) Distribution of overall change (Δ) in displayed parameter (standardized) among cells from control embryos. Horizontal lines on density plots/violins indicate quartiles of distribution. Black circles are individual cells. Statistical comparisons performed by Kolmogorov-Smirnov (KS) test. (X') 2D density plots of standardized variable versus time for all observations/cells in each control embryo in *Figure 2*. Green points are measurements from the representative cells denoted in *Figure 2*. s.d. = standard deviation.

*Figure 3 continued on next page*

*Figure 3 continued*

The online version of this article includes the following figure supplement(s) for figure 3:

**Figure supplement 1.** Endogenous N-cadherin location.

localization during neural tube closure, we imaged the localization of the a GFP-tagged Shroom3 construct lacking the c-terminal Rok-binding domain, similar to a construct previously used to explore Shroom dynamics in *Drosophila* (*Farrell et al., 2017*; *Simões et al., 2014*). In movies of the folding neural plate, the construct localized in a pattern essentially identical to actin, accumulating in both junctional and medial regions of the anterior neural plate (*Figure 4—figure supplement 1*).

Finally, we observed a strikingly different trend in the posterior neural plate, where N-cadherin dynamics did *not* closely parallel actin dynamics. In fact, neither junctional nor medial N-cadherin displayed significant accumulation in the posterior neural plate during the period of observation (*Figure 3D, E*), despite robust actin accumulation in this region (*Figure 3B, C*). Together, these data provide a comprehensive, quantitative description of apical constriction, actin dynamics and N-cadherin localization in the anterior and the posterior neural plate during *Xenopus* neural tube closure. The data further suggest that the mechanisms linking actin and N-cadherin to apical surface area differ in the two regions.

## Mosaic mutation of *Shroom3* reveals distinct anterior and posterior phenotypes in the neural ectoderm

The differences in cell behaviors we observed between anterior and posterior neural ectodermal regions reflect the region-specific nature of NTDs in both humans and animal models. To explore the relationships in more detail, we next turned to loss-of-function manipulation of *shroom3*, which is implicated in human NTDs and has variably penetrant effects on anterior and posterior neural tube closure (*Deshwar et al., 2020*; *Haigo et al., 2003*; *Hildebrand and Soriano, 1999*; *Lemay et al., 2015*).

F0 mutagenesis using CRISPR has recently emerged as a powerful tool in *Xenopus* and zebrafish, and mosaic crispants generated by targeted injections allow simultaneous observation of wild-type and crispant cellular phenotypes so that observations are automatically staged and synchronized (*Aslan et al., 2017*; *Kakebeen et al., 2020*; *Kroll et al., 2021*; *Szenker-Ravi et al., 2018*; *Willsey et al., 2020*). We therefore designed sgRNAs that effectively targeted the coding region of *shroom3*, approximately 28 amino acids from the 5′ end of the transcript, such that any indels generated by CRISPR targeting are likely to disrupt all functional domains of the Shroom3 protein (*Figure 5—figure supplement 1A*).

Using injection into the two dorsal-animal blastomeres at the 8-cell stage to target the neural plate, we demonstrated that our sgRNAs elicited both mutation of the *shroom3* locus as well as the anterior neural tube closure defects expected based on results from knockdown in *Xenopus* using MOs (*Haigo et al., 2003*), as well as the results in mouse genetic mutants (*Hildebrand and Soriano, 1999*). As a critical negative control, injections of sgRNA without Cas9 protein had no effect (*Figure 5—figure supplement 1C, D*).

We next performed more targeted injections to generate mosaic embryos. To do so, we labeled the neural plate by injection of fluorescent reporters into both dorsal blastomeres at the 4-cell stage, and then injected a mixture of *shroom3*-targeted sgRNA, Cas9 protein, and membrane-BFP mRNA into one dorsal blastomere of 8-cell stage embryos (*Figure 5A* and see *Figure 5—figure supplement 1*). We then identified *shroom3* crispant cells via membrane-BFP localization (*Figure 5B*). Because cell junction behavior may be altered at mosaic cell-cell interfaces (i.e. junctions between a control and a crispant cell), we excluded this relatively small number of cells from our analysis. Importantly, this mosaic F0 CRISPR-based approach also generally recapitulated the known phenotype of Shroom3 loss, as we observed gross failure of anterior neural tube closure.

At the level of cell behaviors, we observed a surprising difference in anterior and posterior phenotypes. In the anterior region, *shroom3* crispant cells displayed significantly enlarged apical surfaces at the onset of our imaging (~stage 13), and this phenotype grew more severe over time (*Figure 5C*, left); the majority of cells not only failed to constrict but instead dilated (*Figure 5D*, left). In the posterior, however, the majority of *shroom3* crispant cells still strongly constricted, though collectively they

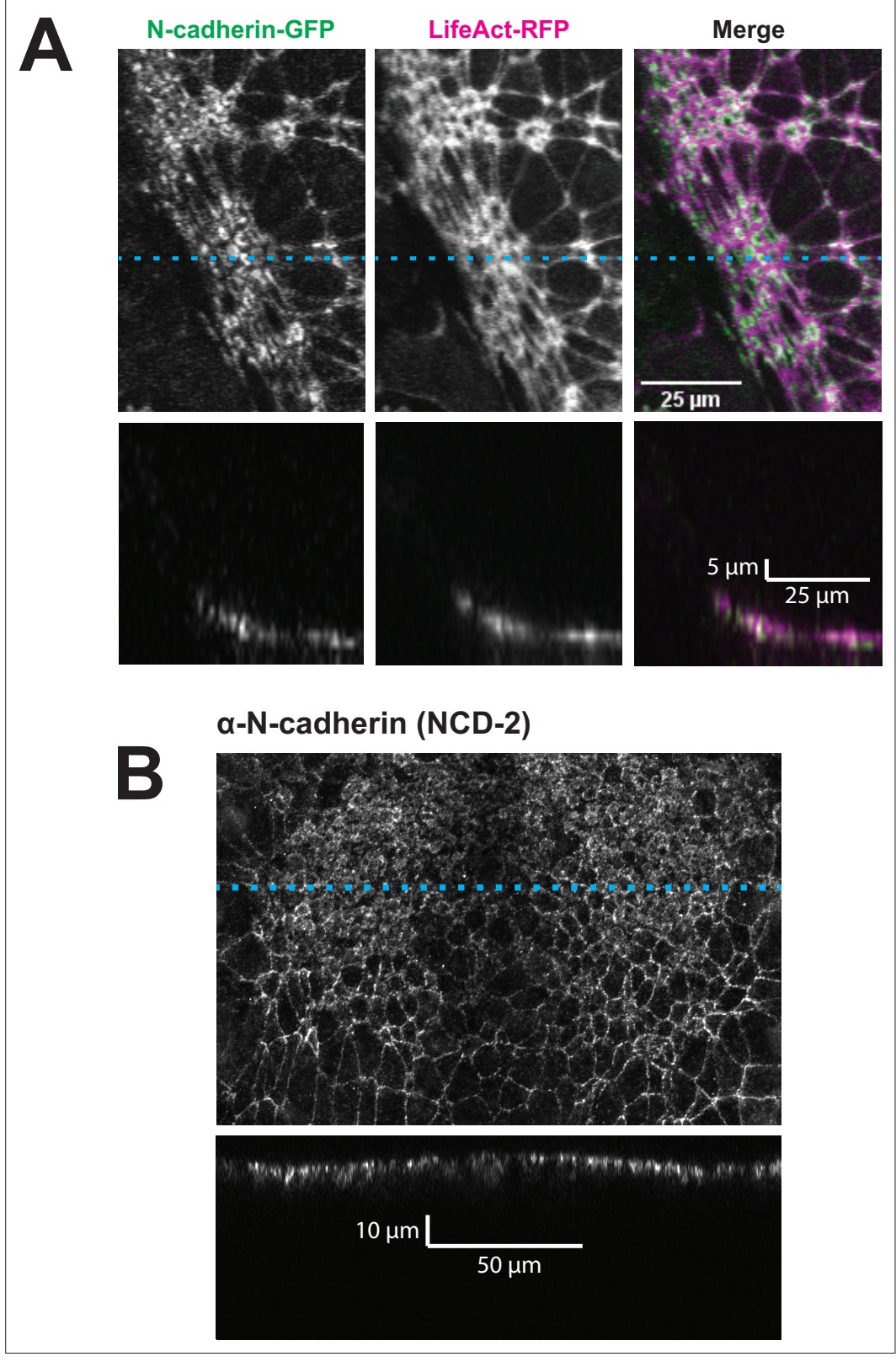

**Figure 4.** N-cadherin localizes both at the apical surface and basally as well. (**A**) XY (top row) and XZ (bottom row) projections of N-cadherin-GFP and LifeAct-RFP in the anterior neural ectoderm of a *Xenopus tropicalis* embryo. (**B**) XY (top panel) and XZ (bottom panel) projections of NCD-2 (monoclonal α-N-cadherin antibody) in the anterior neural ectoderm of a *X. tropicalis* embryo. Dashed cyan lines marks the position of the XZ projection.

*Figure 4 continued on next page*

*Figure 4 continued*

The online version of this article includes the following figure supplement(s) for figure 4:

**Figure supplement 1.** Shroom3ΔC-term colocalizes with LifeAct in the neural ectoderm.

displayed a mildly significant defect in apical constriction (*Figure 5C, D*, right). Thus, the magnitude of apical constriction defects in *shroom3* crispant cells reflects the penetrance of NTDs in the anterior and posterior regions (*Haigo et al., 2003*; *Hildebrand and Soriano, 1999*).

## Loss of Shroom3 uncouples actin dynamics from N-cadherin localization in the anterior neural ectoderm

Loss of Shroom3 disrupts apical actin assembly in the neural plate (*Haigo et al., 2003*; *McGreevy et al., 2015*), but the precise nature of this defect and whether or how it relates to junctional and/or medial actin is unknown. Likewise, N-cadherin is implicated in Shroom3 function and apical constriction (*Lang et al., 2014*; *Nandadasa et al., 2009*; *Plageman et al., 2011b*), but how this relates to actin dynamics is poorly defined. We therefore examined the relationship between apical constriction, actin dynamics, and N-cadherin dynamics, focusing first on the anterior neural plate.

We found that wild-type cells tended to increase both medial and junctional actin localization over time, as expected (*Figure 6A–C*, left). Consistent with the known role in apical actin accumulation (*Haigo et al., 2003*; *Hildebrand, 2005*), *shroom3* crispant cells displayed significantly reduced accumulation of both medial and junctional actin (*Figure 6B, C*, blue violins). However, this effect was surprisingly mild, and in fact, the majority of *shroom3* crispant cells actually *increased* both junctional and medial actin over the period of imaging (*Figure 6B, C*, blue violins).

In contrast to this surprisingly modest change in actin intensity (*Figure 6B, C*), bulk measurements revealed that *shroom3* crispant cells displayed a profound failure to accumulate both junctional and medial N-cadherin, and in fact the majority of cells actually *reduced* N-cadherin levels (*Figure 6A, D, and E*, blue violins). Moreover, this effect was far more pronounced for the medial population of N-cadherin (*Figure 6D*). Thus, loss of Shroom3 elicits a substantially more severe effect on the dynamics of N-cadherin than of actin, apparently uncoupling the two.

## Shroom3 links actin and N-cadherin dynamics to effective apical constriction in the anterior neural ectoderm

To explore these surprising results in more detail, we directly compared changes in apical area with changes in actin and N-cadherin intensity for each cell individually. In 838 control cells, we observed that the vast majority displayed a strong reduction in apical area and a strong increase in both junctional and medial actin intensity (*Figure 7A, B*). As noted in the bulk statistics above, the majority of 147 *shroom3* crispant cells displayed increased actin intensity; however, these crispant cells displayed a bimodal distribution of changes in apical area, with some cells constricting and other cells dilating, yet even cells that increased their apical area after loss of Shroom3 nonetheless accumulated medial and junctional actin (*Figure 7A, B*). Thus, loss of *shroom3* does not lead to *loss* of apical actin in the anterior neural plate but rather to a *reduced accumulation* of apical actin.

N-cadherin localization follows a very different trend, with *shroom3* crispant cells consistently displaying apical surface dilation over time coupled to a strong *reduction* in medial N-cadherin intensity over time (*Figure 7C*). Junctional N-cadherin followed a similar, if less robust, trend (*Figure 7D*). Thus, loss of *shroom3* anteriorly results in a loss of medial and junctional N-cadherin.

An advantage of our large-scale approach is that correlations between parameters provide a more granular view of observed phenotypes. For example, we plotted the correlation between standardized apical area and standardized actin intensity for all cells at all time points (N = ~75 k data points), which revealed a strong negative correlation between apical area and both medial and junctional actin intensity (*Figure 8A, B*). Interestingly, despite the relatively modest impact on bulk actin accumulation (*Figure 6B, C*), loss of *shroom3* effectively abolished the normally strong correlation between apical area and actin intensity (*Figure 8A, B*). Likewise, we found that actin intensity was very strongly positively correlated with N-cadherin intensity in both medial and junctional populations, and this correlation was severely weakened by loss of *shroom3* (*Figure 8C, D*).

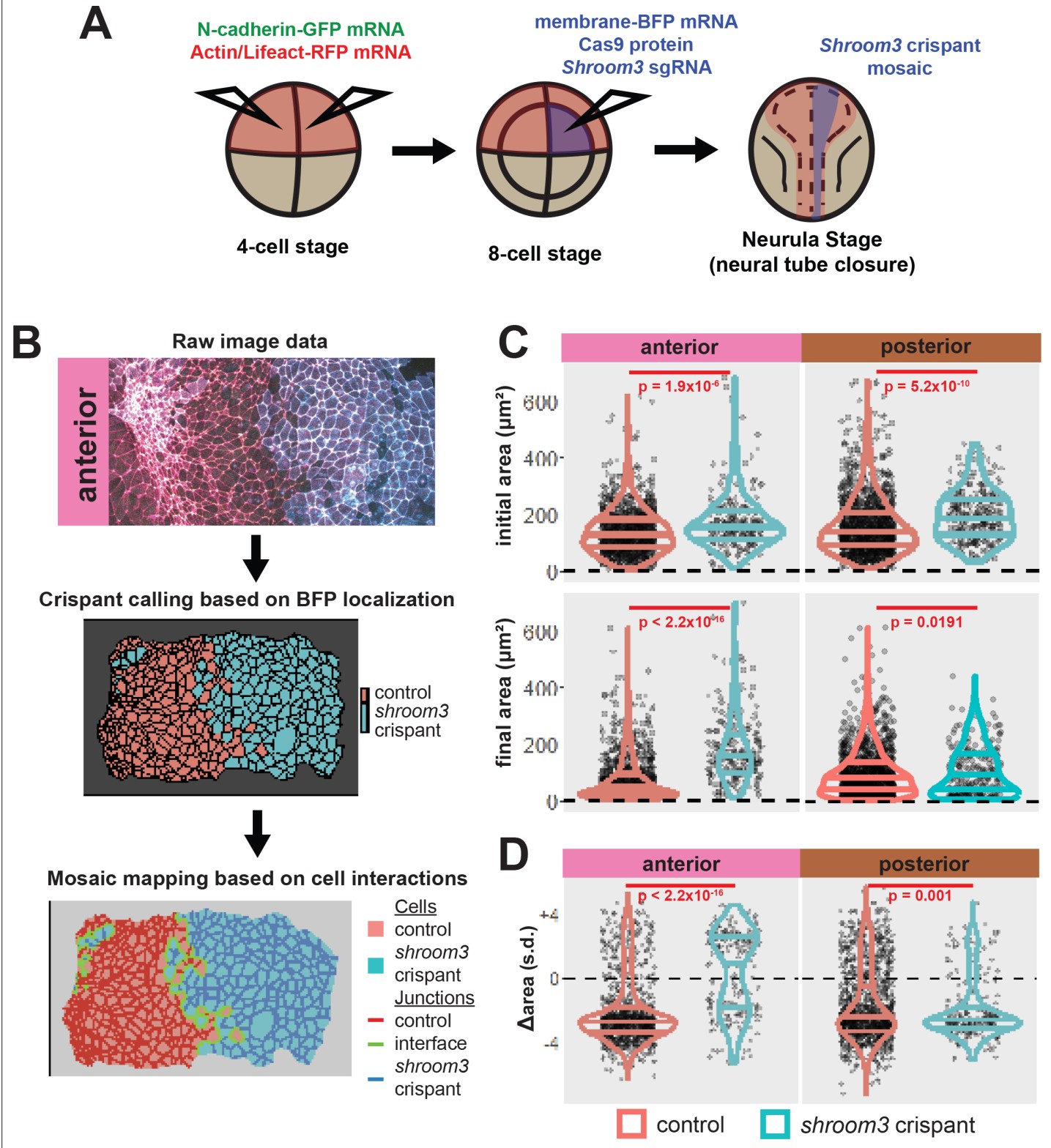

**Figure 5.** Disruption of *shroom3* via mosaic F0 CRISPR mosaic causes differential apical constriction phenotypes between regions of the neural ectoderm. (**A**) Schematic of mosaic F0 CRISPR/Cas9 injections in *Xenopus tropicalis* embryos. (**B**) Workflow of identification and analysis of mosaic F0 crispants. (**C**) Top row, distribution of initial area (square microns) of tracked cells from anterior (left) and posterior (right) embryos. Lower row, distribution of final area (square microns) of tracked cells. (**D**) Distribution of overall change (Δ) in apical area (standardized) from all cells/embryos. In C

*Figure 5 continued on next page*

*Figure 5 continued*

and D, horizontal lines on density plots/violins indicate quartiles of distribution. Black circles are individual cells. Statistical comparisons performed by Kolmogorov-Smirnov (KS) test. Cells situated along the mosaic interface were excluded from these analyses. s.d. = standard deviation.

The online version of this article includes the following figure supplement(s) for figure 5:

**Figure supplement 1.** *shroom3* CRISPR validation.

**Figure supplement 2.** Maps of initial frames and *shroom3* crispant calls for each analyzed embryo.

**Figure supplement 3.** Fluorescent aberration in an anterior-imaged embryo.

Though further exploration of the issue will be required, our large-scale analysis nonetheless generates two interesting hypotheses. First, they suggest an extra-junctional role for medial N-cadherin in apical constriction in the anterior neural plate. Second, they suggest that *shroom3* loss in the anterior neural plate does not prevent actin assembly per se, but rather uncouples medial actomyosin contractility from medial N-cadherin accumulation, and thus uncouples actin assembly from effective apical constriction.

## A distinct mode of action for Shroom3 in the posterior neural plate

As noted above, *shroom3* crispant cells in the posterior neural ectoderm displayed far milder defects in apical constriction over the period of imaging (*Figure 5D*, right). Nonetheless, Shroom3 loss can alone elicit low-penetrance spina bifida and can severely exacerbate spina bifida in combination with certain PCP mutants (*Hildebrand and Soriano, 1999*; *McGreevy et al., 2015*). We therefore explored our image data for insights into this phenotype. We found that both junctional and medial actin accumulated over the course of our imaging in control cells in the posterior neural ectoderm and as we observed anteriorly, both actin populations still increased in *shroom3* crispant cells (*Figure 9A–C*, right). In striking contrast, N-cadherin intensity decreased, both medially and junctionally, in both control and *shroom3* crispant cells (*Figure 9A, D, and E*). Thus, while actin accumulation was very slightly disrupted in posterior *shroom3* cells, N-cadherin dynamics were essentially unaltered in posterior *shroom3* cells.

Considering our data from the anterior neural plate above, these results suggest a fundamentally different relationship between actin, N-cadherin, and apical area in the posterior neural plate. A more granular view of the data revealed two results that reinforced this conclusion. First, the correlations between actin localization and apical area were much weaker in posterior control cells (*Figure 10A, B*) than in anterior control cells (*Figure 8A, B*), suggesting that apical constriction is mechanistically different between the anterior and posterior neural ectoderm. Second, loss of Shroom3 disrupted the actin/area correlation in both regions (*Figures 8A, B and 10A, B*), but this phenotype was much stronger in the anterior region, again consistent with the weaker *shroom3* apical constriction phenotype in the posterior region (*Hildebrand and Soriano, 1999*; *McGreevy et al., 2015*). Finally, while medial and junctional N-cadherin were both strongly negatively correlated with apical area in wild-type anterior cells (*Figure 7F, G*), no correlation whatsoever was observed between N-cadherin and apical area in posterior cells (*Figure 10C, D*).

Together, these results suggest a more complex relationship between actin, N-cadherin, and Shroom3 in the posterior neural plate as compared to the anterior. These results suggest that changes in apical constriction alone are unlikely to explain the role of Shroom3 in posterior NTDs. Instead, the result may reflect the complex interplay of Shroom3-mediated apical constriction and PCP-dependent convergent extension cell behaviors that affect cell-cell junction lengths specifically in the posterior neural plate (e.g. *McGreevy et al., 2015*; *Nishimura et al., 2012*).

## Loss of Shroom3 elicits a subtle but consistent defect in polarized junction shrinking in the posterior neural plate

Our data suggest that changes in apical surface area are unlikely to explain *shroom3*-related posterior NTDs, so turned our attention to the 13,000 individual cell-cell junctions tracked in our dataset. We assessed the behavior of each junction by quantifying the change in length over time, and because junction behaviors during epithelial morphogenesis are frequently polarized with respect to the embryonic axes (*Pinheiro and Bellaïche, 2018*), we next assigned orientations to all junctions in our dataset. Those with a mean orientation less than 45° were designated anteroposterior (AP), as they

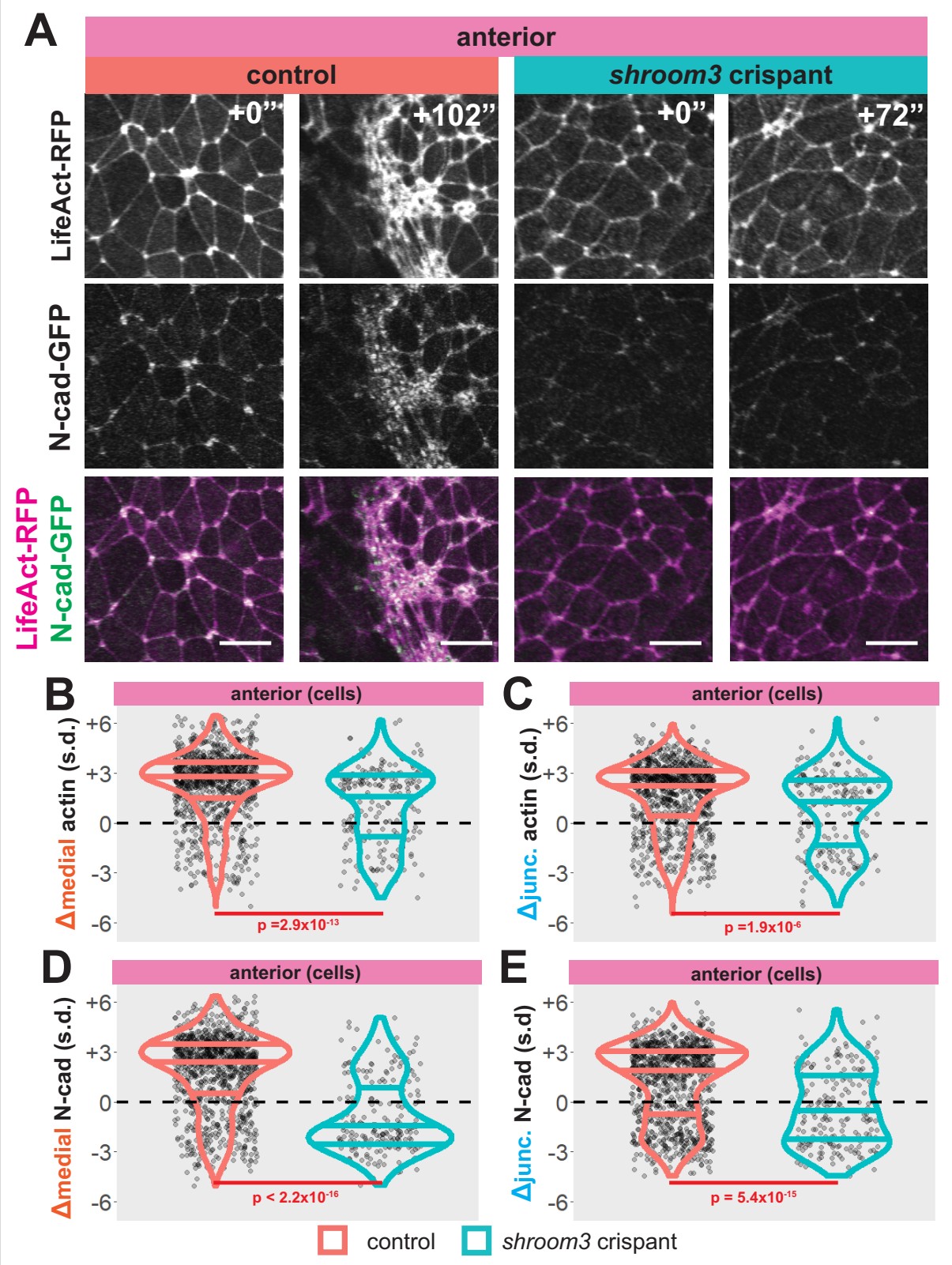

**Figure 6.** Loss of shroom3 disrupts actin and N-cadherin accumulation and constriction in the anterior neural ectoderm. (**A**) Representative images of LifeAct/actin and N-cadherin-GFP (N-cad-GFP) localization in control cells (left) and *shroom3* crispant cells (right) from the anterior region of the neural ectoderm. Scale bar = 15 µm. (**B**) Distribution of overall change (Δ) in medial LifeAct/actin (standardized) from anterior cells. (**C**) Distribution of overall change (Δ) in junctional LifeAct/actin (standardized) from anterior cells. (**D**) Distribution of overall change (Δ) in medial N-cadherin (standardized) from

*Figure 6 continued on next page*

*Figure 6 continued*

anterior cells. (**E**) Distribution of overall change (Δ) in junctional N-cadherin-GFP (standardized) from anterior cells. In B-E, horizontal lines on density plots/violins indicate quartiles of distribution, black circles are individual cells, and statistical comparisons performed by Kolmogorov-Smirnov (KS) test.

represent the anterior face of one cell abutting the posterior face of a neighboring cell; junctions with mean orientations greater than 45° were designated as mediolateral (ML) (*Figure 11A*).

As a positive control for this dataset, we first examined bulk trends. For example, the vast majority of junctions in the anterior neural plate decreased in length, but shortening junctions displayed no bias to their orientation, consistent with the robust, largely isodiametric apical constriction of these cells (*Figure 11—figure supplement 1A, B*). Consistent with our data on apical surface area, anterior *shroom3* crispant cells displayed a robustly significant defect ($p < 2.2 \times 10^{-16}$), with the majority of junctions actually elongating rather than shortening (*Figure 11—figure supplement 1A*, left). A very different trend was observed in the posterior, where most junctions shortened but many elongated (*Figure 11B*, left); shrinking was strongly biased to junction joining AP neighbors, with a reciprocal strong bias for elongation (*Figure 11C*). The converse pattern was observed for actin accumulation, with increasing actin accumulation negatively correlated to decreasing junction length (*Figure 11D*). This pattern is consistent with the known convergent extension cell behaviors in the posterior neural tube and our data here reflect the findings of a previous, smaller scale study of this tissue (*Butler and*

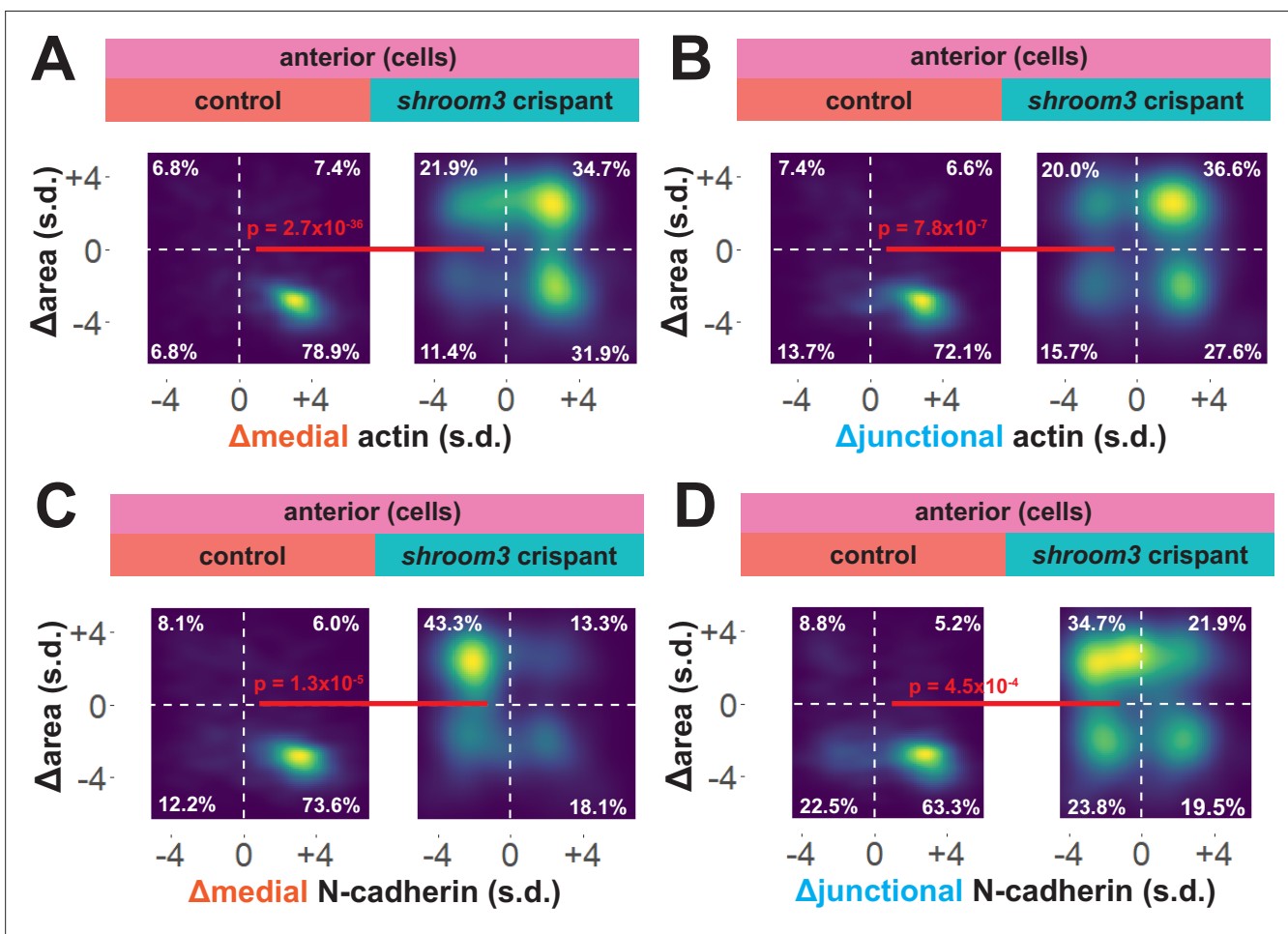

**Figure 7.** Medial N-cadherin accumulation is severely disrupted in anterior *shroom3* crispant cells that fail to apically constrict. (**A-D**) 2D density plots of all observations of apical area versus medial (**A**) or junctional (**B**) LifeAct/actin or medial (**C**) or junctional (**D**) N-cadherin for all cells within each group. Percentages in white indicate the percentage of total cells in each quadrant. Statistical comparisons performed by Peacock test, a 2D implementation of the Kolmogorov-Smirnov (KS) test. Cells situated along the mosaic interface were excluded from these analyses. s.d. = standard deviation.

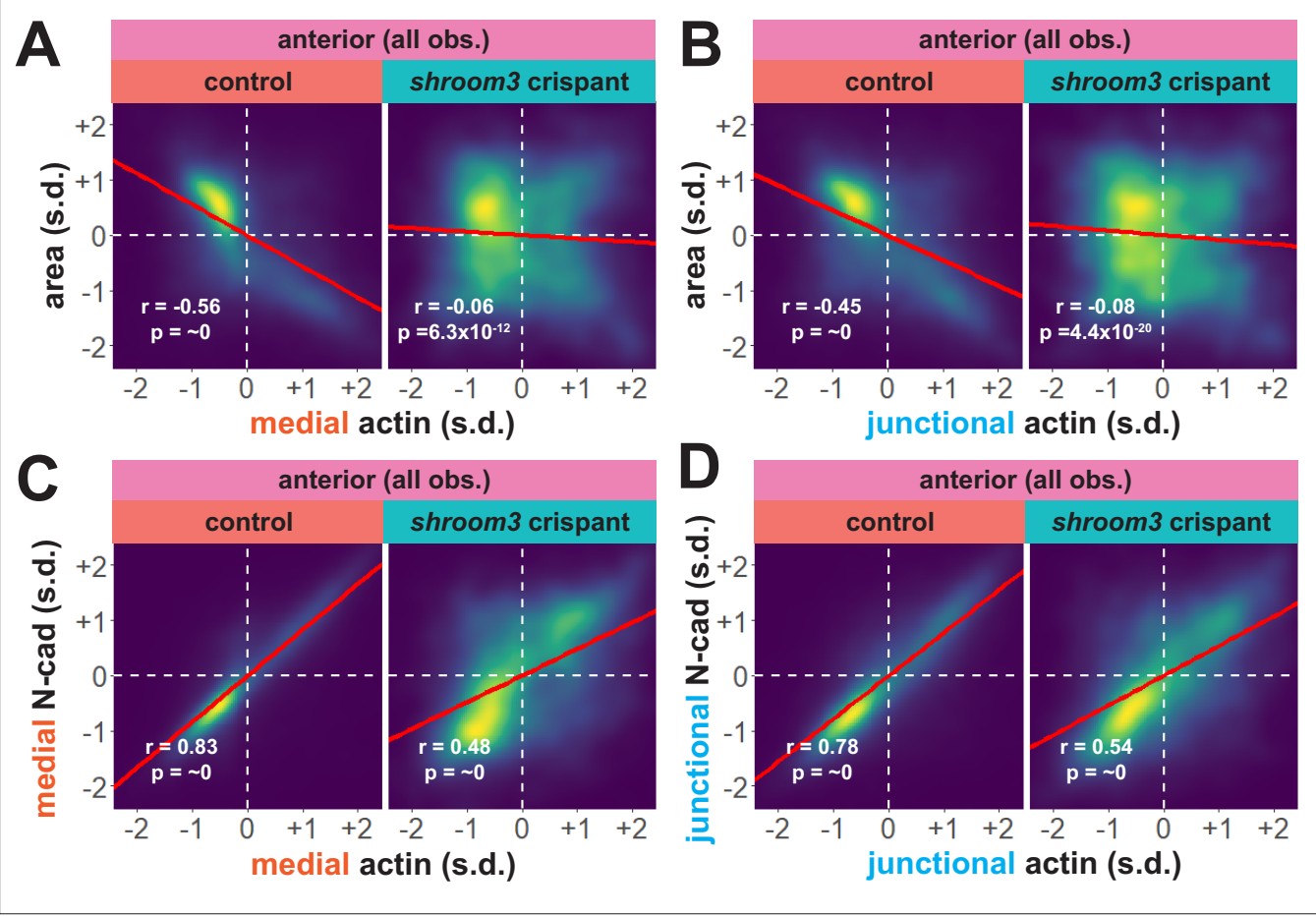

**Figure 8.** Actin and N-cadherin accumulation are uncoupled in anterior *shroom3* crispant cells. (**A and B**) 2D density plots of all observations of medial (**A**) or junctional (**B**) LifeAct/actin versus apical area for all cells within each group. (**C and D**) 2D density plots of all observations of medial (**C**) or junctional (**D**) LifeAct/actin versus N-cadherin at the same domain for all cells within each group. Red lines indicate best-fit line through the observations. Statistics (r and p) are calculated for Pearson's correlation. Cells situated along the mosaic interface were excluded from these analyses. s.d. = standard deviation.

*Wallingford, 2018*) as well as data from fixed mouse embryos (*McGreevy et al., 2015*), providing confidence in the veracity of the dataset.

Our analysis of posterior *shroom3* crispant junctions revealed a barely significant (p = 0.015) change in their bulk behavior (*Figure 11B*, right). This result reflects our cell-level data (*Figure 5D*) and reinforces the conclusion that apical constriction defects are likely not sufficient to explain the posterior NTDs caused by lack of Shroom3. Indeed, a similar modest defect was observed in *shroom3* mutant mice, and that study also revealed a link to the polarization of planar cell arrangement during neural tube closure (*McGreevy et al., 2015*).

One possible explanation for such a polarity phenotype is that Shroom3 is directly required for actomyosin contractions that contribute to junction shortening during CE (see *Huebner et al., 2021*). However, we used Tissue Analyzer to quantify the number of stable neighbor exchanges ('T1 transitions') (*Figure 10F*), and we observed no effect in *shroom3* crispant cells (*Figure 11G*), suggesting that neighbor exchange does not strictly require Shroom3 function. This led us to more carefully explore the effect of Shroom3 loss on polarization of the behaviors of individual junctions.

We found that strongly shrinking junctions were very tightly clustered in the most extreme AP orientation in control cells (i.e. clustered near 0°) (*Figure 11C*, left). By contrast, in *shroom3* crispant cells, shrinking junctions still dominated in the AP quadrant, consistent with previous data from fixed mouse embryos (*McGreevy et al., 2015*), but importantly, they were spread across a wider range of orientations (i.e. more evenly distributed from ~0° to ~40°) (*Figure 11C*, cyan ellipses). Histograms

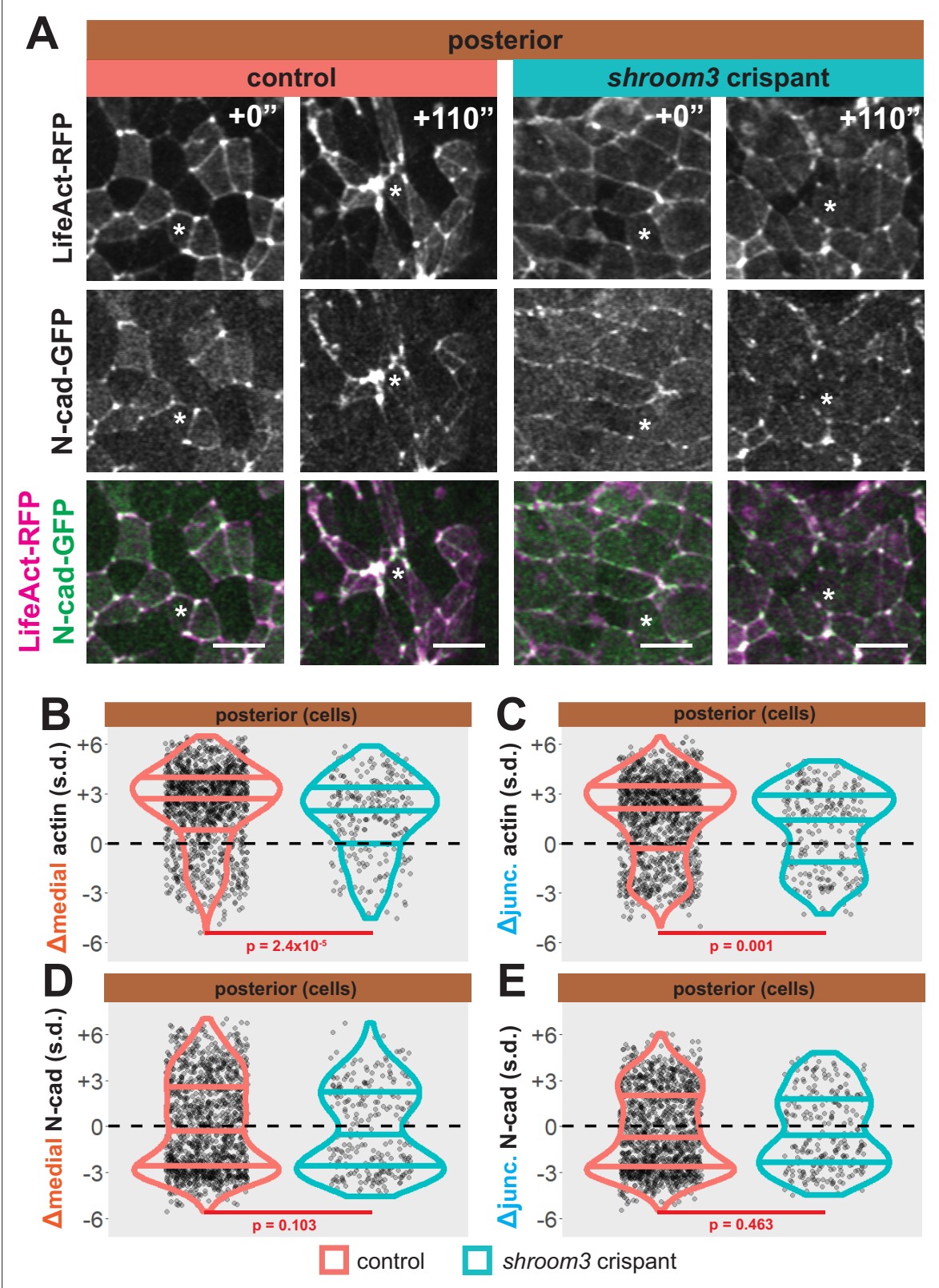

**Figure 9.** Loss of *shroom3* disrupts actin dynamics in the posterior neural ectoderm. (**A**) Representative images of LifeAct/actin and N-cadherin-GFP (N-cad-GFP) localization in control cells (left) and *shroom3* crispant cells (right) from the posterior region of the neural ectoderm. White asterisks mark the same cell in each embryo. Scale bar = 15 μm. (**B**) Distribution of overall change (Δ) in medial LifeAct/actin (standardized) from anterior cells. (**C**) Distribution of overall change (Δ) in junctional LifeAct/actin (standardized) from anterior cells. (**D**) Distribution of overall change (Δ) in medial N-cadherin

*Figure 9 continued on next page*

*Figure 9 continued*

(standardized) from anterior cells. (**E**) Distribution of overall change (Δ) in junctional N-cadherin-GFP (standardized) from anterior cells. In B-E, horizontal lines on density plots/violins indicate quartiles of distribution, black circles are individual cells, and statistical comparisons performed by Kolmogorov-Smirnov (KS) test.

of these data provided a more granular view of this subtle, but significant shift (*Figure 11H*). Elongating junctions were also less polarized in Shroom3 crispants compared to controls (*Figure 11D, H*). Furthermore, actin dynamics very neatly reflected the shrinking and elongating behaviors in both control and *shroom3* crispant cells (*Figure 11D*).

Thus, loss of *shroom3* leads to a subtle but significant shift in the polarization of the junction behaviors that drive convergent extension, a result that could explain the incompletely penetrant posterior NTDs resulting from Shroom3 loss (*Haigo et al., 2003*; *Hildebrand and Soriano, 1999*) and that is very consistent with the known genetic interaction of shroom3 and PCP gene mutations (*McGreevy et al., 2015*). The molecular basis for this interaction remains unclear, but it could relate to the interplay of Shroom3 and N-cadherin. Indeed, we observed a very strong clustering signal for N-cadherin *clearance* from the most polarized elongating junctions (i.e. ~90°) that was not observed for actin in elongating junctions, and this clustering was significantly diminished in *shroom3* crispants (*Figure 11E*).

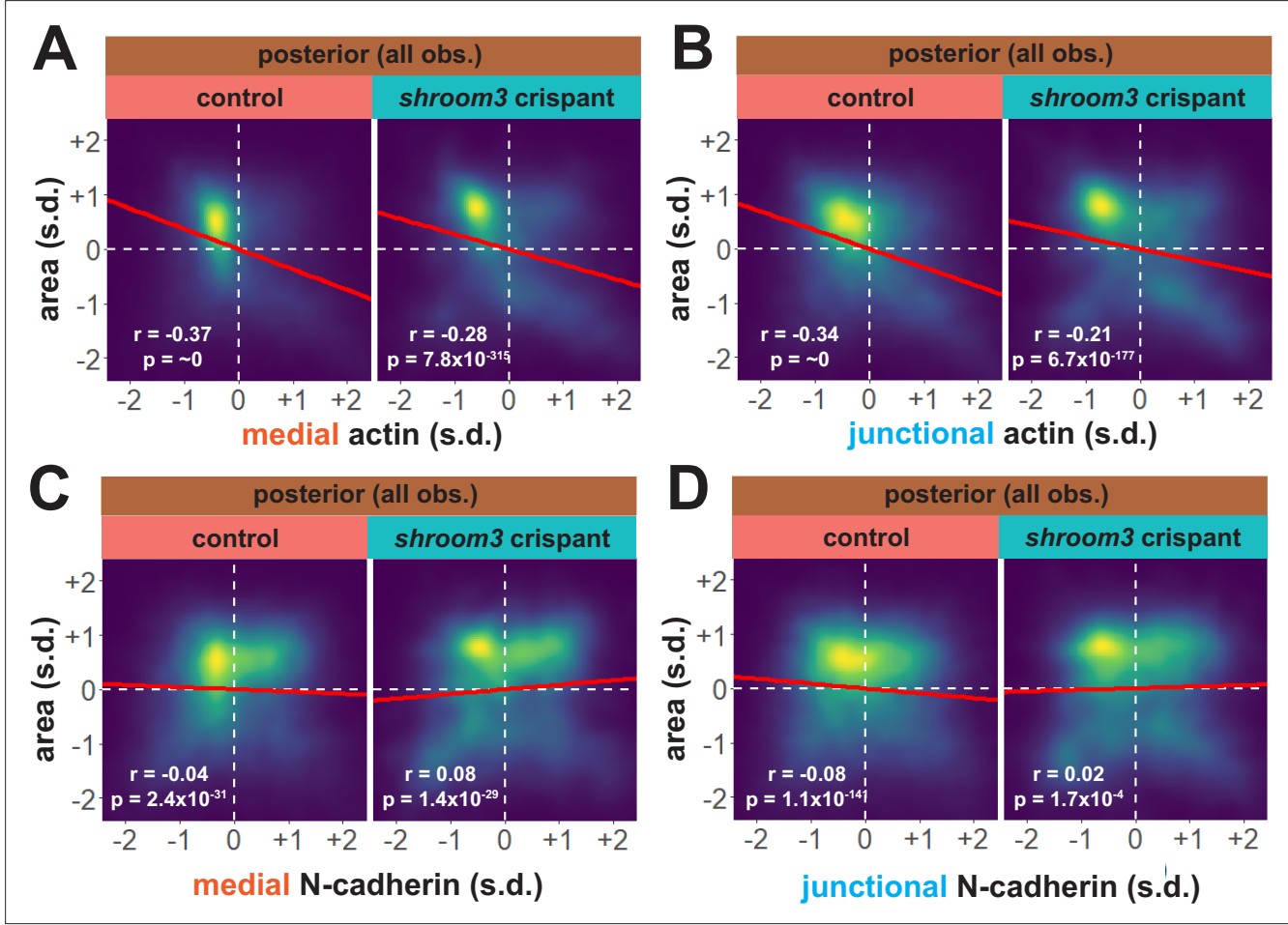

**Figure 10.** Actin and N-cadherin dynamics are highly heterogenous in the posterior neural ectoderm and poorly correlated with apical constriction. (**A and B**) 2D density plots of all observations of apical area versus medial (**A**) or junctional (**B**) LifeAct/actin for all cells within each group. (**C and D**) 2D density plots of all observations of apical area versus medial (**C**) or junctional (**D**) N-cadherin for all cells within each group. Red lines indicate best-fit line through the observations. Statistics (r and p) are calculated for Pearson's correlation. Cells situated along the mosaic interface were excluded from these analyses. s.d. = standard deviation.

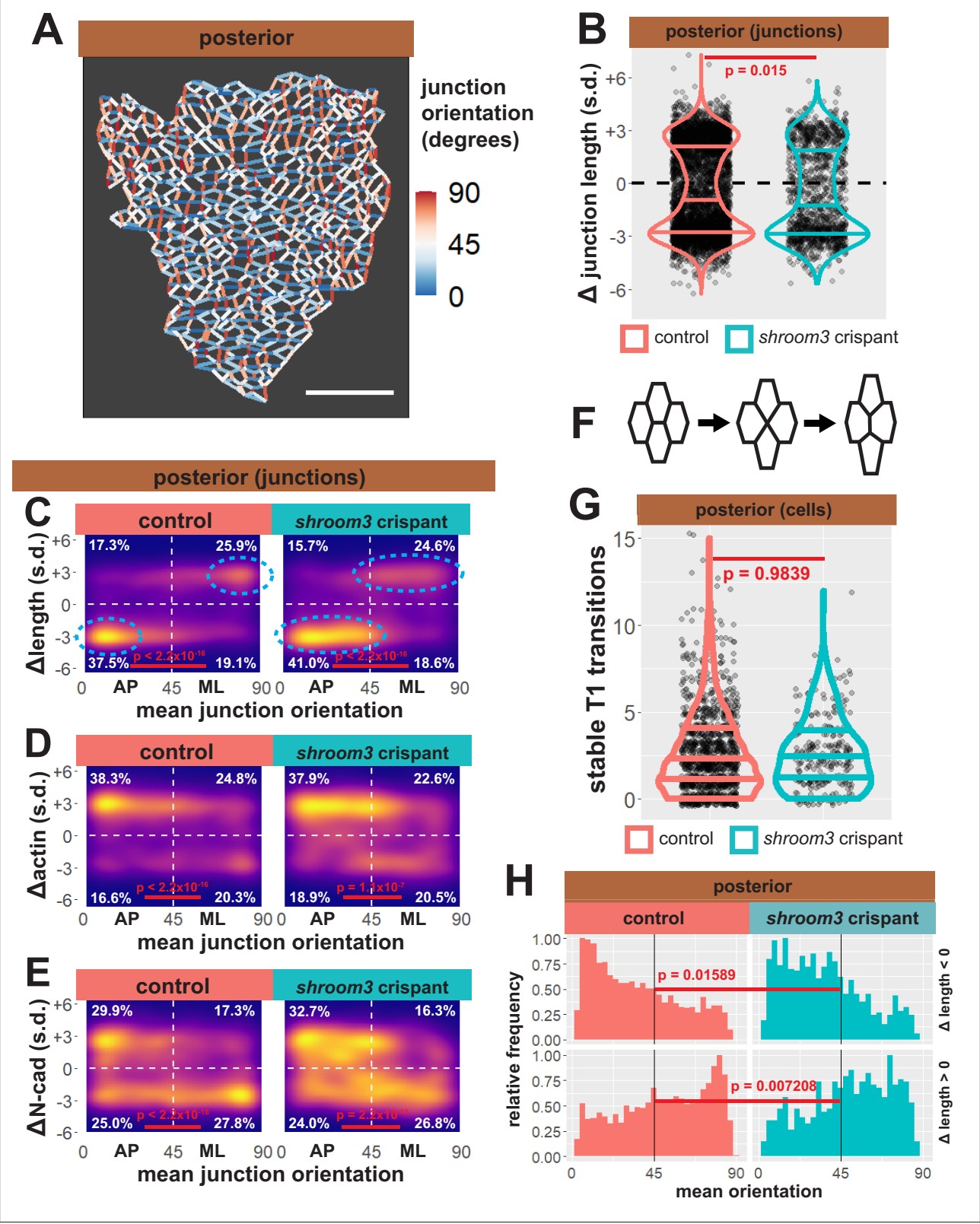

**Figure 11.** Individual junction behaviors are polarized in the posterior neural ectoderm. (**A**) Junction orientation from posterior control embryo from *Figure 2*. Scale bars = 100 μm. (**B**) Distribution of overall change (Δ) in junction length (standardized) from the posterior neural ectoderm. Horizontal lines on density plots/violins indicate quartiles of distribution, black circles are individual cells, and statistical comparisons performed by Kolmogorov-Smirnov (KS) test. (**C**) 2D density plots of all observations of mean junction orientation over time versus overall change (Δ) in junction length

*Figure 11 continued*

(standardized) for all junctions within each group. Dashed cyan ellipses indicate areas of altered polarization between control and shroom3 crispant junctions. (**D**) 2D density plots of all observations of mean junction orientation over time versus overall change (Δ) in junction actin (standardized) for all junctions within each group. (**E**) 2D density plots of all observations of mean junction orientation over time versus overall change (Δ) in junction N-cadherin (standardized) for all junctions within each group. Percentages in white indicate the percentage of total cells in each quadrant. Statistical comparisons performed by Peacock test, a 2D implementation of the KS test. s.d. = standard deviation. (**F**) Example diagram of T1 transition/neighbor exchange within an epithelial tissue. (**G**) Distribution of stable T1 transitions/neighbor exchanges per cell as calculated by Tissue Analyzer. Horizontal lines on density plots/violins indicate quartiles of distribution, black circles are individual cells, and statistical comparisons performed by KS test. (**H**) Histogram showing relative frequencies of mean junction orientation from shrinking (Δ length < 0, upper panels) and growing (Δ length > 0, lower panels) posterior junctions. Compare to 11C. Statistical comparisons performed by KS test.

The online version of this article includes the following figure supplement(s) for figure 11:

**Figure supplement 1.** Individual junction behaviors are mainly anisotropic in the neural ectoderm.

## Conclusions

Understanding the cellular mechanisms contributing to neural tube closure has long challenged embryologists due to the relatively large number of cells involved and the heterogeneity of their behaviors. Indeed, some of the earliest uses of computer simulation in developmental biology focused on understanding the degree to which neural ectoderm cells constrict their apical surfaces during neural tube closure (*Jacobson and Gordon, 1976*). Since then, our understanding of neural tube closure has broadly improved. Hundreds of mutations affecting neural tube closure have been identified (*Harris and Juriloff, 2010*), and the functional interactions between these genes are coming into focus (*McGreevy et al., 2015*; *Murdoch et al., 2014*). However, while time-lapse imaging of single cell behaviors in chicks, frogs, and mice have provided key insights (*Butler and Wallingford, 2018*; *Christodoulou and Skourides, 2015*; *Davidson and Keller, 1999*; *Galea et al., 2017*; *Massarwa and Niswander, 2013*; *Molè et al., 2020*; *Ossipova et al., 2015*; *Pyrgaki et al., 2010*; *Wallingford and Harland, 2002*; *Williams et al., 2014*), our understanding of the cell biology of neural tube closure still lags substantially behind our understanding of the genetics.

Recent research has begun to bridge this gap. In toto imaging has captured neural tube closure as one facet of overall mouse development (*McDole et al., 2018*), and tissue-scale imaging has revealed the cell biological basis of ciliopathy-related NTDs (*Brooks et al., 2020*). In a complement to these recent advancements, we have now used high-resolution, but tissue-scale, time-lapse imaging of both actin and N-cadherin followed by cell-tracking analysis to gain several new insights into the process of apical constriction during neural tube closure.

First, we show that anterior and posterior neural ectoderm cells undergo apical constriction to differing degrees and by different constrictive mechanisms. Second, we show that changes in apical area in the anterior neural ectoderm are more strongly correlated with changes in medial actomyosin localization than junctional actomyosin, indicating that contractility driving apical constriction is more likely to be generated at the medial cell surface. Our results on medial actin localization are largely consistent with smaller scale studies in both *Xenopus* and mice (*Brown and García-García, 2018*; *Christodoulou and Skourides, 2015*; *Suzuki et al., 2017*) and moreover reflect mechanisms described in more detail in the context of *Drosophila* and *C. elegans* gastrulation (*Martin et al., 2009*; *Roh-Johnson et al., 2012*). Conversely, our data show that both medial and junctional actin localization have similar correlations with apical area in the posterior neural ectoderm, suggesting actomyosin contractility may be more balanced across the medial and junctional domains in this region.

Third, we show that N-cadherin accumulates at the medial surfaces of constricting cells in the anterior neural ectoderm, but that N-cadherin localization poorly correlates with apical constriction in the posterior. Interestingly, our analysis suggests that anterior *shroom3* crispant cells fail to constrict less from a lack of actomyosin accumulation and more from an inability to accumulate N-cadherin at the medial surface of cells. This is consistent with a previously reported genetic interaction between *shroom3* and *N-cadherin* (*Lang et al., 2014*; *Li et al., 2021*; *Plageman et al., 2011b*), and suggests that a Shroom3-N-cadherin pathway may drive anterior apical constriction during neural tube closure. The function of this N-cadherin remains to be determined, but two possibilities are suggested by prior studies. First, non-junctional N-cadherin can drive micropinocytosis (*Sabatini et al., 2011*) in cultured cells, so medial N-cadherin may contribute to the known, Shroom3-dependent endocytosis of the constricting apical surface of neuroepithelial cells (*Kowalczyk et al., 2021*; *Lee and Harland, 2010*).

Alternatively, medial N-cadherin might act in a manner similar to non-junctional E-cadherin, forming cis-clusters and directing the assembly of the cortical actin cytoskeleton (*Wu et al., 2015*). Future experiments should be illuminating.

Finally, the study of developmental biology in the 21st century has been marked by explosive increases in the size of experiments – advances in proteomics and transcriptomics are providing an unprecedented depth to our understanding of the molecular workings of cells in embryos. Thus, it is of special importance that we use advances in imaging and data analysis to ask what those cells *actually do and how they do it*. It's notable then that the approach described here can be applied to essentially any protein for which reliable fluorescent reporters are available. Many key questions immediately arise from our work, for example, the localization and dynamics of other crucial players such as Myosin II or Rho kinase. In the posterior neural plate expanding previous smaller scale analyses of PCP protein localization (e.g. *Butler and Wallingford, 2018*) will be of great interest. Finally, the large-scale approach here provides a lower-resolution, but tissue-scale complement to super-resolution imaging advances that are now revealing the subcellular organization of the machinery of morphogenesis (e.g. *Huebner et al., 2021*). Ultimately, then, the work presented here is significant for providing quantitative insights at tissue scale into the interplay of gene function, protein localization, and cell behavior during a biomedically important process in vertebrate embryogenesis.

## Materials and methods

### Animals
Wild-type *X. tropicalis* frogs were obtained from the National *Xenopus* Resource, Woods Hole, MA (*Horb et al., 2019*).

### Injections
Wild-type *X. tropicalis* eggs were fertilized in vitro using sperm from wild-type *X. tropicalis* males using standard methods (*Wlizla et al., 2018*).

*X. tropicalis* embryos were moved to 1/9× MMR +2% Ficoll, then injected in both dorsal blastomeres at the 4-cell stage with 50 pg LifeAct-RFP mRNA and 45 pg *Xenopus* N-cadherin-GFP mRNA, or 80 pg GFP-Shroom3ΔC-term mRNA. In CRISPR-injected embryos, after the next division to reach 8-cell stage, one dorsal blastomere was injected with 1 ng Cas9 protein (PNA Bio), 250 pg *shroom3*-targeted sgRNA (target sequence GUAGCCGGAGAGAUCACUUG, Synthego) (*Figure 5—figure supplement 1A*), and 60 pg membrane(CAAX)-BFP mRNA.

### Antibody staining
*X. tropicalis* embryos were collected at NF stages 13–17 and devitellinized. Embryos were then fixed in 4% paraformaldehyde in PBS for 30 min at room temperature. Embryos were then washed in PBS + 0.01% Triton X-100 3 times for 20 minutes at room temperature. Fixed embryos were blocked in normal goat serum then stained with monoclonal NCD-2 antibody (*Hatta et al., 1987*; *Hatta and Takeichi, 1986*). NCD-2 was diluted 1:10 from supernatant provided by the Developmental Studies Hybridoma Bank, created by the NICHD of the NIH and maintained at The University of Iowa, Department of Biology, Iowa City, IA. Embryos were again washed in PBS + Triton then incubated with a 1:1000 dilution of Invitrogen Alexa Fluor 488 goat α-rat secondary antibody prior to imaging.

### CRISPR genotyping
To test the efficacy of our CRISPR injections in vivo, we injected wild-type *X. tropicalis* with the above-described Cas9 + sgRNA combination in the following cells and stages: 2× injections into 1-cell stage embryos, 1× injections into each blastomere of 2-cell stage embryos, 1× injections into two blastomeres of the 4-cell stage embryo, and 1× injections into all blastomeres of 4-cell stage embryos (*Figure 5—figure supplement 1B*). Uninjected embryos that did not receive any Cas9 + sgRNA were used as controls.

Embryos were allowed to grow to approximately NF stage 40 and then were subjected to whole-embryo DNA extraction. PCR products spanning the *shroom3* target site were generated from each embryo and separated by capillary electrophoresis for fragment analysis. Fragment analysis data was analyzed in R using the *Fragman* package (*Covarrubias-Pazaran et al., 2016*).

Uninjected embryos did not have any indel products at the *shroom3* locus and thus produced one sharp peak at 431 base pairs, corresponding to the size of the wild-type *shroom3* PCR product (*Figure 5—figure supplement 1B*). By contrast, CRISPR-injected embryos returned little to no PCR products at this size (*Figure 5—figure supplement 1B*, dashed red line), indicating that the *shroom3* target site was being efficiently cut by Cas9 and repaired by error-prone pathways. The exception to this were the embryos of which only two blastomeres at the 4-cell stage were injected with Cas9 + sgRNA; as expected, fragments of the wild-type size were detected that theoretically correspond to the uninjected blastomere lineages (*Figure 5—figure supplement 1B*).

Overall, these results indicate that our Cas9 + sgRNA combination efficiently cleaves the *shroom3* target site in vivo. However, our F0 mosaic crispants generated by CRISPR injection into one blastomere at the 8-cell stage do not generate enough crispant cells to be detected by whole embryo PCR at later stages of embryonic development.

## Imaging

Injected embryos were held at 25°C until they reached Nieuwkoop and Faber (NF) stage 12.5. At NF stage 12.5, vitelline envelopes were removed from embryos and embryos were allowed to 'relax' for 30 min. Embryos were then mounted in imaging chambers and positioned for imaging of either the anterior or posterior neural plate.

Embryos were imaged on a Nikon A1R confocal microscope using the resonant scanner. Image quality, Z-stacking, and XY tiling were optimized to generate optimal 3D images of the neural plate at a rate of 1 frame per minute. Ultimately, movies of nine embryos were of sufficient length and quality for analysis, tissue geometry of the initial frame of each of these embryos is presented in *Figure 5—figure supplement 2*.

## Image analysis

Raw 3D images were projected to 2D via maximum intensity and underwent initial segmentation of cell boundaries using the FIJI plugin Tissue Analyzer (*Aigouy et al., 2010*; *Aigouy et al., 2016*). The segmentation of an initial frame was hand-corrected, and this hand-corrected segmentation was used to train a classifier using the programs CSML and EPySEG (*Aigouy et al., 2020*; *Ota et al., 2018*). CSML and EPySEG were used to generate segmentation for subsequent frames, which were then further hand-corrected in Tissue Analyzer.

After hand-correction, Tissue Analyzer was used to track both cell surfaces and cell junctions, then generate a database of measurements of size and fluorescent intensities for each cell and junction over time. Values for medial and junctional localization of imaged markers in cells were calculated as average pixel fluorescence intensity across the entirety of each respective domain (i.e. total fluorescence of a region divided by the area of the region). Similarly, localization of imaged markers to individual junctions was calculated as an average across the entire junction.

For individual junctions, errors in junction length caused by Z-displacement and projection were corrected in Matlab.

Tissue Analyzer databases were imported to R and further analyzed and manipulated primarily using the *tidyverse* package (*Wickham et al., 2019*).

## Data analysis

Cell tracks shorter than 30 frames and junction tracks shorter than 15 frames were discarded.

Individual cell and junction tracks were smoothed by averaging over a 7 frames/min window (*Figure 1—figure supplement 1A, B*). Individual cell tracks were further mean-centered and standardized so that variables are measured in standard deviations rather than fluorescence or size units (*Figure 1—figure supplement 1C*). This standardization allows us to analyze dynamics of cell size and protein localization across a population of cells while controlling for initial size and fluorescence of cells. In an example embryo, cells begin and end tracking with a variety of apical surface areas (*Figure 1—figure supplement 1D'*), but once the cell tracks are mean-centered and standardized it becomes clear that the cells are behaving similarly at a population level (*Figure 1—figure supplement 1D"*).

Embryo 'b' had a fluorescence anomaly during imaging that resulted in a reduction in overall observed fluorescence followed by a recovery (*Figure 5—figure supplement 3A, B*). Cells were

tracked through the anomaly (*Figure 5—figure supplement 3C*), but fluorescent values for the frames 23–45 were discarded (*Figure 5—figure supplement 3*, red dashed box).

Cells were determined to be wild-type versus *shroom3* crispant by a membrane-BFP localization threshold specific to each embryo (*Figure 5B*, middle panel). Crispant calls were then manually annotated in cases along the mosaic interface where thresholding produced crispant calls deemed incorrect.

Individual junctions were determined to be wild-type versus *shroom3* crispant versus mosaic interface based on the status of the cells the junction was situated between. Wild-type junctions are situated between two wild-type cells, *shroom3* crispant junctions are situated between two *shroom3* crispant cells, and mosaic interface junctions are situated between a wild-type and a *shroom3* crispant cell (*Figure 5B*, lower panel).

Junction orientations were corrected so that the ML axis of the embryo was set at 0° and the AP axis of the embryo was set at 90° (*Figure 11B*).

## Cell data parameters

**cell_surfaces** – frame-by-frame tracked data for cells
*region*: relative region of the neural ectoderm, that is, 'anterior' or 'posterior'
*movie:* label for each individual embryo analyzed
*track_id_cells*: cell tracking label that is unique to cells within an embryo but may be repeated between different embryos
*minute*: time per each individual embryo/movie in minutes
*center_x_cells*: pixel X coordinate of centroid of each cell in each frame
*center_y_cells*: pixel Y coordinate of centroid of each cell in each frame
*vx_coords_cells*: pixel XY coordinates of the vertices of each cell in each frame in X:Y#X:Y format
*CRISPR*: CRISPR status of each cell, that is, 'control' or '*shroom3* crispant', called based on membrane-BFP localization.
*control_neighbors*: number of cell neighbors that are 'control' in each frame
*crispant_neighbors*: number of cell neighbors that are '*shroom3* crispant' in each frame
*apical_area_pixels*: cell apical area in pixels (measured within a one pixel constriction of the segmented cell junctions)
*apical_area_pixels_smoothed: apical_area_pixels* averaged over 7 frames, –3 and +3 frame frame in question
*apical_area_micron_smoothed: apical_area_pixels_smoothed* converted to square microns
*apical_area_standardized: apical_area_pixels_smoothed* mean-centered and scaled per cell track (via R 'scale' function), measured in standard deviations
*medial_actin*: mean LifeAct-RFP fluorescent intensity at the medial apical domain of each cell in arbitrary units (measured within a one pixel constriction of the segmented cell junctions)
*medial_actin_smoothed: medial_actin* averaged over 7 frames, –3 and +3 frame frame in question
*medial_actin_standardized: medial_actin_smoothed* mean-centered and scaled per cell track (via R 'scale' function), measured in standard deviations
*junctional_actin*: mean LifeAct-RFP fluorescent intensity at the junctional domain of each cell in arbitrary units (measured at segmented cell junctions)
*junctional_actin_smoothed: junctional_actin* averaged over 7 frames, –3 and +3 frame frame in question
*junctional_actin_standarized: junctional_actin_smoothed* mean-centered and scaled per cell track (via R 'scale' function), measured in standard deviations
*medial_Ncadherin*: mean N-cadherin-GFP fluorescent intensity at the medial apical domain of each cell in arbitrary units (measured within a one pixel constriction of the segmented cell junctions)
*medial_Ncadherin_smoothed: medial_Ncadherin* averaged over 7 frames, –3 and +3 frame frame in question
*medial_Ncadherin_standarized: medial_Ncadherin_smoothed* mean-centered and scaled per cell track (via R 'scale' function), measured in standard deviations

*junctional_Ncadherin*: mean N-cadherin-GFP fluorescent intensity at the junctional domain of each cell in arbitrary units (measured at segmented cell junctions)

*junctional_Ncadherin_smoothed: junctional_Ncadherin* averaged over 7 frames, –3 and +3 frame frame in question

*junctional_Ncadherin_standarized: junctional_Ncadherin_smoothed* mean-centered and scaled per cell track (via R 'scale' function), measured in standard deviations

*medial_memBFP*: mean membrane(CAAX)-BFP fluorescent intensity at the medial apical domain of each cell in arbitrary units (measured within a one pixel constriction of the segmented cell junctions)

*medial_memBFP_smoothed: medial_memBFP* averaged over 7 frames, –3 and +3 frame frame in question

*junctional_memBFP*: mean membrane(CAAX)-BFP fluorescent intensity at the junctional domain of each cell in arbitrary units (measured at segmented cell junctions)

*junctional_ memBFP_smoothed: junctional_ memBFP* averaged over 7 frames, –3 and +3 frame frame in question

**cell_surface_stats** – summary statistics for cells

*region*: relative region of the neural ectoderm, that is, 'anterior' or 'posterior'

*movie:* label for each individual embryo analyzed

*track_id_cells*: cell tracking label that is unique to cells within an embryo but may be repeated between different embryos

*CRISPR*: CRISPR status of each cell, that is, 'control' or '*shroom3* crispant', called based on membrane-BFP localization.

*at_mosaic_interface*: TRUE denotes that cell was next to another cell of the other *CRISPR* type at some point during tracking, that is, control cells next to *shroom3* crispant cells and vice versa. FALSE indicates that a cell was only next to cells of the same CRISPR type, that is, control cells next to only other control cells. Cells labeled TRUE were not used in quantitative analyses in this paper.

*start_area_micron*: initial apical area of a cell in square microns (calculated from *apical_area_micron_smoothed)*

*end_area_micron*: final apical area of a cell in square microns (calculated from *apical_area_micron_smoothed)*

*delta_apical_area*: final value of *apical_area_standardized* minus initial value of *apical_area_ standardized* (measured in standard deviations/s.d.)

*delta_medial_actin*: final value of *medial_actin_standardized* minus initial value of *medial_actin_ standardized* (measured in standard deviations/s.d.)

*delta_junctional_actin*: final value of *junctional_actin_standardized* minus initial value of *junctional_actin_standardized* (measured in standard deviations/s.d.)

*delta_medial_Ncadherin*: final value of *medial_Ncadherin_standardized* minus initial value of *medial_Ncadherin_standardized* (measured in standard deviations/s.d.)

*delta_junctional_Ncadherin*: final value of *junctional_Ncadherin_standarized* minus initial value of *junctional_Ncadherin_standarized* (measured in standard deviations/s.d.)

**junctions** – per minute/junction measurements of junctions

*region*: relative region of the neural ectoderm, that is, 'anterior' or 'posterior' *movie:* label for each individual embryo analyzed

*track_id_junctions*: junction tracking label that is unique to junctions within an embryo but may be repeated between different embryos. Junctions are determined as interactions between cells.

*CRISPR*: CRISPR status of each junction based on the cells the junction is between, that is, 'control' is between two control cells, '*shroom3* crispant' is between two *shroom3* crispant cells, and 'at mosaic interface' is between a control cell and a *shroom3* crispant cell. Junctions 'at mosaic interface' were not included in quantitative analyses in this paper.

*minute*: time per each individual embryo/movie in minutes

*vx_1_x, vx_1_y:* pixel XY coordinates of first vertex of junction

*vx_2_x, vx_2_y:* pixel XY coordinates of second vertex of junction

*actin*: mean LifeAct-RFP fluorescent intensity across the junction, measured in arbitrary units

*actin_smooth:*

*actin* averaged over 7 frames, –3 and +3 frame frame in question

*actin_standardized: actin_smooth* mean-centered and scaled per cell track (via R 'scale' function), measured in standard deviations

*Ncadherin*: N-cadherin-GFP fluorescent intensity across the junction, measured in arbitrary units

*Ncadherin_smooth: Ncadherin* averaged over 7 frames, –3 and +3 frame frame in question

*Ncadherin_standardized: Ncadherin_smooth* mean-centered and scaled per cell track (via R 'scale' function), measured in standard deviations

*length_px*: length of junction in maximum intensity projection measured in pixels

*length_micron: length_px* converted to microns

*delta_z_micron*: Z-distance between first and second vertex of junction, in microns. Calculated by determining where the maximum intensity projection sampled the Z-stack for each vertex in the LifeAct-RFP channel.

*length_corrected: length_micron* corrected for z-distance between vertices using *delta_z_micron*, measured in microns.

*length_smooth: length_corrected* averaged over 7 frames, –3 and +3 frame frame in question

*length_standardized: length_smooth* mean-centered and scaled per cell track (via R 'scale' function), measured in standard deviations

*orientation*: orientation of junction relative to AP axis in degrees, where 0 is aligned with ML axis and 90 is aligned with AP axis.

**junction_stats** – summary statistics for junctions *region*: relative region of the neural ectoderm, that is, 'anterior' or 'posterior'

*movie:* label for each individual embryo analyzed

*track_id_junctions*: junction tracking label that is unique to junctions within an embryo but may be repeated between different embryos. Junctions are determined as interactions between cells.

*CRISPR*: CRISPR status of each junction based on the cells the junction is between, that is, 'control' is between two control cells, '*shroom3* crispant' is between two *shroom3* crispant cells, and 'at mosaic interface' is between a control cell and a *shroom3* crispant cell. Junctions 'at mosaic interface' were not included in quantitative analyses in this paper.

*delta_length*: final value of *length_standardized* minus initial value of *length_standardized* (measured in standard deviations/s.d.)

*delta_actin*: final value of *actin_standardized* minus initial value of *actin_standardized* (measured in standard deviations/s.d.)

*delta_Ncadherin*: final value of *Ncadherin_standardized* minus initial value of *Ncadherin_standardized* (measured in standard deviations/s.d.)

*mean_orientation*: average of *orientation* over junction track

## Acknowledgements

Special thanks to Pavak Shah and Claire McWhite for assistance with coding and data analysis and to the Wallingford lab for manuscript comments. This work was funded by NICHD Ruth L Kirschstein NRSA F32 HD094521 for AB and R01HD099191 to JW.

## Additional information

### Funding

| Funder | Grant reference number | Author |
| --- | --- | --- |
| Eunice Kennedy Shriver National Institute of Child Health and Human Development | R01HD099191 | John B Wallingford |

The funders had no role in study design, data collection and interpretation, or the decision to submit the work for publication.

### Author contributions

Austin T Baldwin, Conceptualization, Data curation, Formal analysis, Funding acquisition, Investigation, Supervision, Writing - original draft, Writing - review and editing; Juliana H Kim, Data curation, Formal analysis, Investigation, Software; Hyemin Seo, Investigation, Methodology, Visualization; John B Wallingford, Conceptualization, Funding acquisition, Project administration, Supervision, Writing - original draft, Writing - review and editing

### Author ORCIDs

Austin T Baldwin (iD) http://orcid.org/0000-0002-6099-0873
Juliana H Kim (iD) http://orcid.org/0000-0001-6634-4525
John B Wallingford (iD) http://orcid.org/0000-0002-6280-8625

### Ethics

Approved by IACUC at UT austin: AUP-2021-00167, Expiration date: 08/16/2024.

### Decision letter and Author response

Decision letter https://doi.org/10.7554/eLife.66704.sa1
Author response https://doi.org/10.7554/eLife.66704.sa2

## Additional files

### Supplementary files

• Transparent reporting form

### Data availability

We have deposited two types of files on Dryad: 'cell_surfaces' and 'junctions' are spreadsheets containing the frame-by-frame measurements of cell/junctions size, location, fluorescent protein localization, and other parameters. 'cell_surface_stats' and 'junction_stats' are spreadsheets of summary statistics generated from the frame-by-frame data describing overall changes in parameters in individual cells and junctions. These data can be downloaded at: https://doi.org/10.5061/dryad.zw3r2289b.

The following dataset was generated:

| Author(s) | Year | Dataset title | Dataset URL | Database and Identifier |
| --- | --- | --- | --- | --- |
| Baldwin AY, Kim J, Seo H, Wallingford J | 2022 | Global analysis of cell behavior and protein localization dynamics reveals region-specific functions for Shroom3 and N-cadherin during neural tube closure | https://dx.doi.org/10.5061/dryad.zw3r2289b | Dryad Digital Repository, 10.5061/dryad.zw3r2289b |

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
