## [Editor Report]

This manuscript by Baldwin and colleagues on vertebrate neural tube closure will be of interest to developmental and cell biologists studying tissue morphogenesis as well as human geneticists focusing on neural tube defects. It is timely, as it introduces a new technology for large-scale imaging of cell behaviours in large embryos. Specifically, it uses advanced image analysis to quantitatively describe and correlate active cell behaviours and localization dynamics of key cytoskeletal and adhesion proteins driving a central step of neural tube closure. Data analysis is detailed and followed by careful conclusions.

---

## [Decision Letter]

**Decision letter after peer review:**

Thank you for submitting your article "Global analysis of cell behavior and protein localization dynamics reveals region-specific functions for *Shroom3* and N-cadherin during neural tube closure" for consideration by *eLife*. Your article has been reviewed by 3 peer reviewers, including Elke Ober as Reviewing Editor and Reviewer #1, and the evaluation has been overseen by a Reviewing Editor and Marianne Bronner as the Senior Editor. The following individuals involved in review of your submission have agreed to reveal their identity: Α Yap (Reviewer #2); Jeff Hildebrand (Reviewer #3).

Essential revisions:

1) The authors need to revisit their conclusions, which in places are overextended. For example, it seems too much to conclude that "the primary effect of shroom 3 in the anterior neural plate is …on the coupling of medial actomyosin …with medial N-cadherin accumulation" (p 10, bottom). It is the most obvious change in their current data set, but that data set seems too limited to call it the "primary effect".

Similarly, the manuscript strength is the new analytical approach of molecular and cellular changes at tissue scale rather than in increasing the understanding of neural tube morphogenesis. Therefore, we suggest to better frame the manuscript around this new analytical approach and its capacity to yield data of great richness for understanding morphogenetic processes in vertebrates.

Addressing the points raised below will support the potential of the approach for gaining mechanistic insights and overall strengthen the manuscript.

2) Estimating molecular concentration: Please clarify and address whether the molecular concentration of analyzed proteins has been corrected for change in size of the analyzed area over time. For example, the medial N-cadherin increases with apical constriction (Figure 6), but is this because the apical surface is getting smaller (i.e. same amount of apical N-cadherin, but now more concentrated) or is there an increase in the total amount of protein in the apical surface?

3) Clarify the analysis of medial signals: How do the authors correct for noise in their analyses? Lifeact can also bind G-actin and some of the punctate N-cadherin-GFP could potentially be in vesicles rather than at the apical membrane (see also point 5).

4) The non-junctional distribution of N-cadherin and its dynamic changes during apical constriction represent an exciting result. However, solely using ectopically expressed N-cadherin-GFP to investigate its function is not sufficient, as it may introduce overexpression artifacts. Please provide data corroborating that endogenous N-cadherin behaves similar to the exogenous protein. We appreciate that based on the available reagents (e.g. antibodies) this would likely represent static images. If this proves experimentally not possible, this should at a minimum be addressed in the text.

5) Could the current data be used to assess the trafficking of the N-Cadherin? For instance, can the authors determine if N-cadherin moving from junctional to medial locations, is being trafficked directly to the medial membrane, or being internalized from the medial region, or perhaps some other dynamic behavior. This could help provide more information regarding the mechanistic role of N-cadherin.

6) Does medial N-cadherin co-distribute with Myosin II, ppRLC, or Rho-kinase? Based on previous studies of apical constriction in other model systems this is an attractive assumption. This should be tested experimentally to support and mechanistically corroborate the correlation of molecular events and cell shape changed described in this manuscript. For instance, combine N-cadherin with a MyosinII, a GTP-Rho localization sensor (e.g. Bement lab) or the AHPH system (Piekny and Glotzer, 2008) reporter to generate relevant time-series.

7) Given that non-junctional N-cadherin has been associated with diverse cellular functions apart from adhesion, please discuss its possible role in this current context.

8) Support the functional inactivation of *Shroom3*: The sequence analysis provided is compelling but actually demonstrating reduced protein would strengthen the method.

What proportion of indels are 3 bp or multiple of 3, could resulting in-frame deletions or monoallelic indels explain for instance the 2 populations observed for instance in Fig5D, E?

Moreover, please clarify which particular domain is targeted by the *Shroom3* gRNA employed in this study? How is it expected to impair its function, e.g. complete loss of function, deletion of a specific functional domain? If the latter, could a truncated protein exert partial functions? How would this effect interactions with actin or N-cadherin and relate to the specific phenotypes observed?

9) *Shroom3* spatial expression: How is *Shroom3* expressed throughout the extent of the anteroposterior neural epithelium, given that it seems to exert different effects in the anterior and posterior parts? Likewise, would be important to see the *Shroom3* subcellular distribution in the neural plate to determine if there is a population of *Shroom3* in medial positions analogous to N-cadherin.

10) Clarification of sample numbers and data integration of different samples: The samples included in this study and presented in Methods Appendix2 display apparent differences, therefore additional information is required for the number of samples that contribute to each analysis and how data were compared and/or integrated. For instance, the anterior samples show differences in cell size and asymmetries within the tissue. Please explain the reason for this and how is this accounted for when comparing quantifications between samples. This should include how staging between samples was achieved, and the related registration allowing comparison of resulting cell behaviors.

Related to this, there seem to be only two posterior samples containing *Shroom3* crispant cells, please describe the variability between samples, similar to above. If only two samples were interrogated, a third sample needs to be included. In general, the sample number per experiment should be greater than two.

11) Describing the data analysis. Please introduce in the Results a few sentences that explain the "standardization" approach that are used to present the data. While this is in the Methods and Appendix, the approach is not one commonly seen, and it would be good to orient readers less familiar with the approach.

12) Facilitating the interpretation of some the 2D density plots in figure 10. Consider an alternative way or simplification of the graphs without breaking the data out into several additional graphs. If difficult, a more detailed description in the methods should be helpful to the reader and included.

13) The 2D density plots in figure 6D and E have such sharp edges; they look artificial. Please check and address whether this is just an issue with the PDF, thresholding, or some other technical issue.

14) Discuss limitations inherent to the approach, including: (i) a relatively limited number of molecular parameters are interrogated (F-actin and N-cadherin). It is possible, for example, that changes in contractility which drive junctional shortening (relevant for the analysis in Figure 10) are due to changes in actin organization (that may not be readily captured by overall measures of quantity) or activity of Myosin II (which is not measured here). Given the scale of the experiments that are involved, it would be technically challenging to interrogate more molecular players at the same time, representing a potential limitation.

(ii) the dynamics are relatively coarse-grained. For example, changes in cadherin levels that occur over hours may not capture changes in molecular turnover.

*Reviewer #1:*

In this manuscript Baldwin and colleagues investigate a key morphogenetic step of neural tube closure, namely the apical constriction of the neural epithelium. Using *Xenopus* as a model for live imaging, they monitor the underlying cell behaviours over time, including large-scale quantitative analyses of cell shape changes and subcellular localization of key proteins, actin and N-cadherin. This revealed clear differences in the anterior and posterior neural plate with respect to apical constriction dynamics and cell junction dynamics. The cytoskeletal interactor *Shroom3* mediates aspects of these cell behaviours in a region-specific fashion, as revealed by F0 CRISPR/Cas9 mutagenesis. *Shroom3* seems to control partially similar actin and N-cadherin localization, however, its loss leads to stronger defects only in anterior neural tube closure. The strength of this study is the highly quantitative nature of the approach: quantifying the morphogenetic behaviours of individual cells at the tissue level, which leads to an integration of previous observations with new insights into region-specific differences of cell behaviours and their correlation with characteristic actin and N-cadherin subcellular localisation. The very interesting observations however require further substantiation by functional experiments, as well as clarification of some experimental details.

The non-junctional distribution of N-cadherin and its dynamic changes during apical constriction represent an exciting result. Given that non-junctional N-cadherin has been associated with diverse functions, please clarify its possible role in this context. If it would be possible to determine its region-specific function(s) without interfering with cell adhesion, would add to the mechanistic understanding.

Which particular domain is targeted by the *Shroom3* gRNA employed in this study? How is it expected to impair its function, e.g. complete loss of function, deletion of a specific functional domain? If the latter, could a truncated protein exert partial functions? How would this effect interactions with actin or N-cadherin and relate to the specific phenotypes observed?

What proportion of indels are 3 bp or multiple of 3, could this or monoallelic indels explain for instance the 2 populations observed in Figure 5?

There seem to be two populations of n-cadherin positive cells in the posterior domain, how are they spatially distributed and would this affect morphogenesis on the tissue level?

How is *Shroom3* expressed throughout the extent of the anteroposterior neural epithelium? Given that *Shroom3* seems to have different effects in the anterior and posterior neural epithelium, information about its spatial distribution and sub cellular localisation would be informative for the interpretation of the results.

The samples included in this study and presented in Methods Appendix2 display apparent differences, therefore additional information is required for the number of samples that contribute to each analysis and how data were compared. For instance, the anterior samples show differences in cell size and asymmetries within the tissue. Please explain the reason for this and how is this accounted for when comparing quantifications between samples?

Along the same lines, there seem to be only two posterior samples containing *Shroom3* crispant cells, please describe the variability between samples. If there is variability, a third sample needs to be included in the study.

How was the staging between samples and adjustment between resulting cell behaviours achieved? For instance, there seem to be fewer cells/lower signal at the time of rapid constriction in the posterior neural plate.

*Reviewer #2:*

Morphogenesis entails cell-level behaviours that generate tissue-level (multicellular) consequences. The challenge of characterizing and dissecting those cellular behaviours in the context of a tissue or organism is made even greater because (a) regional variation is often critical for morphogenetic processes (e.g. convergent extension, tube closure); and (b) cellular level activity is likely to have stochastic components that make it difficult to identify regional patterns of variation. Traditional approaches in cell biology or in the molecular genetic dissection of development have been less well-equipped to deal with these complexities.

To address this issue, Baldwin et al., describe an analysis pipeline that utilizes live optical imaging to capture cellular-level events within a morphogenetically-active tissue. Combining tiling and confocal imaging with large scale image analysis, the authors characterize patterns of cell shape change, subcellular F-actin and N-cadherin accumulation, in the neural tube of *Xenopus tropicalis*. This addresses the challenge of capturing both cell- and tissue-level behaviours in a relatively large organism. They reveal a regional diversity of behaviours between the anterior and posterior neural tubes, which are also disparately affected when Shroom 3, a known regulator of neural tube development, is depleted.

I think that the major contribution of this paper is to introduce an approach to the large-scale imaging of tissues in a "big" vertebrate. The value is that the approach can extract cellular-level data, combined with regional information at the tissue level. And it can generate large numbers for good statistics. Although this has been done for smaller organisms, less progress has been made for larger samples. Another attraction of the approach is that the authors have used readily-available imaging technology and analysis tools. To be clear, I think that this is a valuable contribution for the field.

I am less certain of how much this paper represents in terms of new "biological" insight. The principal experimental intervention is *Shroom3* KO and this changes some of the patterns of correlation (between cell shape change, cadherin concentration and F-actin). But the current analysis doesn't help me better understand which of those changes responsible for the abnormal morphogenesis of the Shroom 3 phenotype. This is for the following reasons (which are inherent features of the approach):

i) A relatively limited number of molecular parameters are interrogated (F-actin and N-cadherin). It is possible, for example, that changes in contractility which drive junctional shortening (relevant for the analysis in Figure 10) are due to changes in actin organization (that may not be readily captured by overall measures of quantity) or activity of Myosin II (which is not measured here). Given the scale of the experiments that are involved, it would be technically challenging to interrogate more molecular players at the same time, but this is a potential limitation that should be acknowledged.

ii) The dynamics are relatively coarse-grained. For example, changes in cadherin levels that occur over hours may not capture changes in molecular turnover.

So, while I think that the paper will make a contribution that could be suitable for *eLife*, I would ask the authors to consider the following issues.

Specific

1) I think that the authors sometimes draw their bow(s) a little too long. For example, it seems too much to conclude that "the primary effect of shroom 3 in the anterior neural plate is …on the coupling of medial actomyosin …with medial N-cadherin accumulation" (p 10, bottom). It is the most obvious change in their current data set, but that data set seems too limited to call it the "primary effect".

2) Medial signals. How do the authors correct for noise in their analyses? Lifeact can also bind G-actin and some of the punctate N-cadherin-GFP could potentially be in vesicles rather than at the apical membrane.

3) Estimating molecular concentration. I may have missed this, but have the authors also done an analysis where they correct for change in size of the region analysed. For example, the medial N-cadherin increases with apical constriction (Figure 6), but is this because the apical surface is getting smaller (i.e. same amount of apical N-cadherin, but now more concentrated) or is there an increase in the total amount of protein in the apical surface?

4) Focus of the paper. The authors seemed to frame this manuscript as an effort to better understand the morphogenesis of the neural tube. But, as noted above, from that perspective I was disappointed. I think the paper may be better framed as a new analytical approach, which yields data of great richness for future work.

*Reviewer #3:*

This manuscript by Baldwin, Kim, and Wallingford describes an in-depth, robust analysis of cell morphology, actin dynamics, and N-cadherin distribution in the neural plate of *Xenopus tropicalis*. This analysis further describes the role of *Shroom3* in the regulation or participation in these processes. This work represents an important study of the cellular dynamics associated with neural tube formation and identifies some fundamental differences between the anterior and posterior regions of the neural plate that may contribute to birth defects that occur in different regions of the neural tube. The work presented should have an impact in the areas of cell and tissue morphogenesis, methods for analysis of complex cell dynamics in the context of intact tissues/embryos, epithelial organization, and neural tube formation.

Strengths:

Overall, the conclusions and interpretation are well supported by the detailed analysis of cellular and cytoskeletal dynamics during neural plate morphogenesis. It provides unique and valuable insights into the differences in cellular behavior in the anterior and posterior regions of the neural plate. This approach and level of analysis can help to set a standard for the combined use of Crispr, live cell imaging, and computational analysis to assess the function of specific proteins in dynamic processes in whole embryos.

Weaknesses:

My major concern is with the statement that the data presented advance our understanding of the mechanistic role for actin, *Shroom3*, N-cadherin, and contractility in apical constriction. This study shows beautiful and compelling correlative data that there is a, actin-*Shroom3*-N-cadherin network or pathway (or some other type of "interaction") that is important. However, this does not seem to provide a mechanism beyond what has previously been shown in other studies. This, however, does not invalidate the significance of the work presented.

I think some of the points below might help to provide more mechanistic insights, if possible.

1. I have a concern with only using ectopically expressed N-cadherin-GFP to investigate it's function. If possible, it would be useful to have some data indicating that the endogenous N-cadherin behaves in a way that is similar to the exogenous protein. I appreciate these would likely have to be static images and this may not be possible based on the available reagents. At a minimum, it should be addressed in the text if not possible to address experimentally.

2. Is it possible, using the data that has already been collected, to assess the trafficking of the N-Cadherin? For instance, can the authors determine if the N-cadherin is moving from junctional to the medial locations, being trafficked directly to the medial membrane, is it being internalized from the medial region, or perhaps some other dynamic behavior. This could help provide more information regarding the mechanistic role of N-cadherin.

3. Along these same lines, do the authors know if the medial N-cadherin co-distributes with Myosin II, ppRLC, or Rho-kinase? It might be presumed to the case based on previous studies of apical constriction in other model systems, but it would be interesting to see if this is the case.

4. Similarly, it might also be useful to see the *Shroom3* distribution in the neural plate to determine if there is a population of *Shroom3* in medial positions analogous to N-cadherin. I think the sequence analysis provided is compelling but actually demonstrating reduced protein would strengthen the method.

Additional comments:

I had a difficult time interpreting some the 2D density plots in figure 10. I'm not sure if there is really a better way to simplify it without breaking the data out into several additional graphs. Perhaps a more detailed description in the methods would be helpful to the reader.

I'm curious why the 2D density plots in figure 6D and E have such dramatic edges? These look artificial and I wonder if this is just an issue with the PDF, thresholding, or some other technical issue.

---

## [Author Response]

Essential revisions:1) The authors need to revisit their conclusions, which in places are overextended. For example, it seems too much to conclude that "the primary effect of shroom 3 in the anterior neural plate is …on the coupling of medial actomyosin …with medial N-cadherin accumulation" (p 10, bottom). It is the most obvious change in their current data set, but that data set seems too limited to call it the "primary effect".Similarly, the manuscript strength is the new analytical approach of molecular and cellular changes at tissue scale rather than in increasing the understanding of neural tube morphogenesis. Therefore, we suggest to better frame the manuscript around this new analytical approach and its capacity to yield data of great richness for understanding morphogenetic processes in vertebrates.Addressing the points raised below will support the potential of the approach for gaining mechanistic insights and overall strengthen the manuscript.

We have re-written much of the manuscript, focusing on the technical advances here, but also clarifying the distinctions between our findings and our conclusions, which in most cases we have softened.

Furthermore, we have now removed data that we deemed superfluous, thereby shortening and simplifying the manuscript.

2) Estimating molecular concentration: Please clarify and address whether the molecular concentration of analyzed proteins has been corrected for change in size of the analyzed area over time. For example, the medial N-cadherin increases with apical constriction (Figure 6), but is this because the apical surface is getting smaller (i.e. same amount of apical N-cadherin, but now more concentrated) or is there an increase in the total amount of protein in the apical surface?

Yes, we report changes in the *mean* pixel intensity normalized for area. This has been clarified in the revision on lines 147-149.

3) Clarify the analysis of medial signals: How do the authors correct for noise in their analyses? Lifeact can also bind G-actin and some of the punctate N-cadherin-GFP could potentially be in vesicles rather than at the apical membrane (see also point 5).

To account for "noise" within our data, we have smoothed the data by averaging the data in individual cell tracks over a 7-frame/minute window. We have included a description of this in the main text (lines 149151) and included a diagram in Figure 1 figure supplement 1. As LifeAct cannot strictly distinguish between F-actin and G-actin, we have not distinguished between F-actin and G-actin in the manuscript. We now explicitly describe what we observed and our interpretation thereof. We now also report on the apicobasal position of N-cad relative to actin (lines 197-200, Figure 4) and we discuss the possible mechanisms of action for medial N-cad (Discussion, lines 448-462).

4) The non-junctional distribution of N-cadherin and its dynamic changes during apical constriction represent an exciting result. However, solely using ectopically expressed N-cadherin-GFP to investigate its function is not sufficient, as it may introduce overexpression artifacts. Please provide data corroborating that endogenous N-cadherin behaves similar to the exogenous protein. We appreciate that based on the available reagents (e.g. antibodies) this would likely represent static images. If this proves experimentally not possible, this should at a minimum be addressed in the text.

We agree this is crucial. We therefore examined this issue using immunostaining for endogenous Ncadherin. These experiments, presented in the new Figure 3 figure supplement 1 and lines 194-197 confirm our findings with N-cadherin-GFP.

5) Could the current data be used to assess the trafficking of the N-Cadherin? For instance, can the authors determine if N-cadherin moving from junctional to medial locations, is being trafficked directly to the medial membrane, or being internalized from the medial region, or perhaps some other dynamic behavior. This could help provide more information regarding the mechanistic role of N-cadherin.

We regret that we do not have enough time resolution (1 frame per minute) to resolve the directionality of N-cadherin movement. However, we have shown some new images from our data that provide some insight into the relative apicobasal positioning of actin and N-cadherin (Figure 4), and we discuss the possibility of N-cad endocytosis (lines 197-203 and 448-462).

6) Does medial N-cadherin co-distribute with Myosin II, ppRLC, or Rho-kinase? Based on previous studies of apical constriction in other model systems this is an attractive assumption. This should be tested experimentally to support and mechanistically corroborate the correlation of molecular events and cell shape changed described in this manuscript. For instance, combine N-cadherin with a MyosinII, a GTP-Rho localization sensor (e.g. Bement lab) or the AHPH system (Piekny and Glotzer, 2008) reporter to generate relevant time-series.

We agree that these are interesting experiments, but as the reviewers themselves point out in Point 14 (below), examining these additional markers is extremely daunting given the scale of each experiment here. Thus, we followed the advice in Point 14 and now clearly discuss this limitation of our approach and leave analysis of other markers for a future paper.

7) Given that non-junctional N-cadherin has been associated with diverse cellular functions apart from adhesion, please discuss its possible role in this current context.

The possible roles for medial N-cad are now discussed in lines 448-462 of the discussion.

8) Support the functional inactivation of Shroom3: The sequence analysis provided is compelling but actually demonstrating reduced protein would strengthen the method.What proportion of indels are 3 bp or multiple of 3, could resulting in-frame deletions or monoallelic indels explain for instance the 2 populations observed for instance in Fig5D, E?

Unfortunately, there are no available antibodies that detect *Shroom3* protein in *Xenopus* (mouse antibodies do not cross-react). We hasten to add, however, that all aspects of the phenotype we observe with *shroom3* CRISPR recapitulate known phenotypes not only of *shroom3* morphants in *Xenopus* but also of dominant-negative *shroom3* in *Xenopus* and genetic mutants in mice. This explicitly stated now in the manuscript, and coupled to our sequence analysis, we hope these findings will satisfy the reviewers.

Regarding the proportions of indels and their relationship to phenotypes, we have no way of exploring this possibility. That said, as we now clarify in the revision on lines 236-239 and in Figure 5 figure supplement 1A, our sgRNA targets amino acid ~28 of the ~3000 amino acid *Shroom3* protein, making it unlikely that distinct change-of-function mutations could be introduced. We therefore feel it more appropriate not to speculate on the issue.

Moreover, please clarify which particular domain is targeted by the Shroom3 gRNA employed in this study? How is it expected to impair its function, e.g. complete loss of function, deletion of a specific functional domain? If the latter, could a truncated protein exert partial functions? How would this effect interactions with actin or N-cadherin and relate to the specific phenotypes observed?

As noted in point 8, above, the sgRNA targets amino acid ~28 of the ~3000 amino acid *Shroom3* protein. This is substantially N-terminal to all defined domains in the protein. This is now stated on lines 236-239 and diagrammed in Figure 5 figure supplement 1A.

9) Shroom3 spatial expression: How is Shroom3 expressed throughout the extent of the anteroposterior neural epithelium, given that it seems to exert different effects in the anterior and posterior parts? Likewise, would be important to see the Shroom3 subcellular distribution in the neural plate to determine if there is a population of Shroom3 in medial positions analogous to N-cadherin.

*Xenopus shroom3* is expressed along the entire length of the closing *Xenopus* neural plate (Haigo et al., *Current Biology* 2003), as we now indicate on lines 86-88.

As for the protein localization, unfortunately no antibodies that detect *Xenopus Shroom3* are available. Compounding this problem, ectopic expression of wild-type *Shroom3* causes a severe gain-of-function phenotype in early embryos, eliciting strong apical constriction of blastomeres that precludes analysis of *Shroom3*-GFP localization at neural plate stages.

That said, ectopically expressed, tagged *Shroom3* clearly decorates the entire apical surface in diverse epithelial cell types (see Haigo, 2003; Lee et al., 2009), and this point is now made on lines 204-208 of the revision.

Finally, in a more direct attempt to attempt to address this concern, we took a cue from previous work on *Drosophila* Shroom, and we expressed a C-terminal (Rok-binding domain) truncation of *Xenopus Shroom3*.

We found his construct co-accumulates with both medial and junctional actin in the closing neural plate. These data are now discussed on lines 208-214 and shown in Figure 4 figure supplement 1.

10) Clarification of sample numbers and data integration of different samples: The samples included in this study and presented in Methods Appendix2 display apparent differences, therefore additional information is required for the number of samples that contribute to each analysis and how data were compared and/or integrated. For instance, the anterior samples show differences in cell size and asymmetries within the tissue. Please explain the reason for this and how is this accounted for when comparing quantifications between samples. This should include how staging between samples was achieved, and the related registration allowing comparison of resulting cell behaviors.

Differences in cell size and fluorescent intensity arise from many sources: (a) staging, (b) natural variation, (c) variation arising from microinjection of mRNAs, etc. As such, we have focused not so much raw values of size and intensity, but rather in the changes in these values over time. Thanks to our cell tracking paradigm, we are able to mean-center and scale cell size and fluorescence parameters per individual cell track and convert measures of area and fluorescence (i.e. microns and arbitrary units) to standard deviations. Thus both within and between embryos, each cell is analyzed individually for relative changes in parameters over time and then integrated into the overall dataset. We have clarified this in the text on lines 157-164 and is additionally diagrammed in Figure 1 figure supplement 1.

Related to this, there seem to be only two posterior samples containing Shroom3 crispant cells, please describe the variability between samples, similar to above. If only two samples were interrogated, a third sample needs to be included. In general, the sample number per experiment should be greater than two.

We have now included additional videos and analysis.

11) Describing the data analysis. Please introduce in the Results a few sentences that explain the "standardization" approach that are used to present the data. While this is in the Methods and Appendix, the approach is not one commonly seen, and it would be good to orient readers less familiar with the approach.

As described above in response to Point 10, we have improved the description of this method on lines 157164 of the main text and diagrammed the standardization in Figure 1 figure supplement 1.

12) Facilitating the interpretation of some the 2D density plots in figure 10. Consider an alternative way or simplification of the graphs without breaking the data out into several additional graphs. If difficult, a more detailed description in the methods should be helpful to the reader and included.

We have broken out a large part of the data in Figure 10 (now Figure 11) to Figure 11 figure supplement 1 to aid with legibility. These data are now described in more detail and the original plots have now been annotated to guide the reader (Figure 11C, cyan ellipses). In addition, histograms have been extracted from different regions of the plots (Figure 11H) to provide more granular view of specific results.

13) The 2D density plots in figure 6D and E have such sharp edges; they look artificial. Please check and address whether this is just an issue with the PDF, thresholding, or some other technical issue.

This was a technical issue with a background layer on the plots. This background layer has been removed in all density plots in the manuscript.

14) Discuss limitations inherent to the approach, including: (i) a relatively limited number of molecular parameters are interrogated (F-actin and N-cadherin). It is possible, for example, that changes in contractility which drive junctional shortening (relevant for the analysis in Figure 10) are due to changes in actin organization (that may not be readily captured by overall measures of quantity) or activity of Myosin II (which is not measured here). Given the scale of the experiments that are involved, it would be technically challenging to interrogate more molecular players at the same time, representing a potential limitation.(ii) the dynamics are relatively coarse-grained. For example, changes in cadherin levels that occur over hours may not capture changes in molecular turnover.

Limitation of the approach are now discussed on lines 463-477.